# Motion sickness and sense of presence in a virtual reality environment developed for manual wheelchair users, with three different approaches

**Zohreh Salimi**⬥¤*, **Martin William Ferguson-Pell**\*

Department of Rehabilitation Science, Faculty of Rehabilitation Medicine, University of Alberta, Edmonton, Alberta, Canada

¤ Current address: Autism Spectrum Disorder Research Center, Hamedan University of Medical Sciences, Hamedan, Iran
* martin.ferguson-pell@ualberta.ca (MWFP); zsalimi@ualberta.ca (ZS)

**Data Availability Statement:** All datasets are available from the Figshare repository database at: https://doi.org/10.6084/m9.figshare.13669862.v1.

## Abstract

Visually Induced Motion Sickness (VIMS) is a bothersome and sometimes unsafe experience, frequently experienced in Virtual Reality (VR) environments. In this study, the effect of up to four training sessions to decrease VIMS in the VR environment to a minimal level was tested and verified through explicit declarations of all 14 healthy participants that were recruited in this study. Additionally, the Motion Sickness Assessment Questionnaire (MSAQ) was used at the end of each training session to measure responses to different aspects of VIMS. Total, gastrointestinal, and central motion sickness were shown to decrease significantly by the last training session, compared to the first one. After acclimatizing to motion sickness, participants' sense of presence and the level of their motion sickness in the VR environment were assessed while actuating three novel and sophisticated VR systems. They performed up to four trials of the Illinois agility test in the VR systems and the real world, then completed MSAQ and Igroup Presence Questionnaire (IPQ) at the end of each session. Following acclimatization, the three VR systems generated relatively little motion sickness and high virtual presence scores, with no statistically meaningful difference among them for either MSAQ or IPQ. Also, it was shown that presence has a significant negative correlation with VIMS.

## Introduction

The Virtual Reality (VR) industry is rapidly growing and finding applications in widely different areas. Many believe that VR will have a prominent role in many aspects of life, including business, education, entertainment, medicine, and research facilities, but there are others who disagree. One of the main reasons is the Visually Induced Motion Sickness (VIMS) that is associated with VR [1–3], particularly the immersive VRs, e.g. head-mounted displays [4]. VIMS is an unpleasant and nauseating experience that if experienced, can have a strong role in

**Funding:** The authors received no specific funding for this work. Canadian Foundation for Innovation (CFI 30852 Leading Edge Fund 2012) infrastructure grant provided the capital equipment for the project (received by MFP). The funders had no role in study design, data collection and analysis, decision to publish, or preparation of the manuscript.

**Competing interests:** The authors have declared that no competing interests exist.

throwing VR users' sense of presence off and subsequently, adversely affecting how they behave and perform in VR [4]; so much so that the VR users may be reluctant to use VRs again. VIMS is an accumulative [5] construct during a VR session that when triggered can rapidly escalate and the consequences such as disorientation and vertigo may remain for hours [6] or even days [7]. This, in addition to potentially shrinking the pool of potential VR users, could be unsafe, e.g. when the users return to normal activities, as reported in the literature [8].

To figure out what factors influence motion sickness in VR, Chattha et al. [9] conducted an experiment with 46 participants. The VR system used in this study was the Oculus Rift DK2. The factors studied were gender, genre (horror/pleasant), prior VR experience, and prior motion sickness experience. All participants first tried the pleasant genre and then the horror genre, and their heart rate, sugar level, and blood pressure were recorded before the experiment, between the two genres, and after the experiment. Participants also completed a motion sickness questionnaire after finishing each genre. Authors reported that the horror genre made all 46 participants sick, more severely for women, when compared to the pleasant genre. This motion sickness severity was shown both in questionnaire results and in elevated heart rate and blood pressure, as well as decreased sugar level. They conclude these results indicate that fear is related to motion sickness. They also reported that prior experience of VR and 3D games, or prior motion sickness experience do not impact motion sickness.

Some factors are understood to have roles in triggering VIMS, such as eye separation, field of view, frame rate, latency [1], interactivity [10], quality of the images and projection, and calibration of devices. However, even after optimizing these factors, VIMS still happens, as a consequence of a mismatch between vestibular and visual stimuli [11]. There have been ways identified to deal with this issue, but clearly, in many VR applications, there will always be an inconsistency between the two. Fortunately, VIMS, and motion sickness in general, tend to diminish when subjects are exposed to them during multiple sessions repeated over days [3, 11–14]. Also, other techniques could be used such as Puma exercises [15] and vestibular training [14] to help reduce motion sickness.

As mentioned, when there is a mismatch between vestibular and visual stimuli, VIMS could trigger. This is roughly whenever the user moves around, or looks around, in the virtual world. We also mentioned that VR has a potential to be used to assist with many different areas, including for training and conducting related research studies for wheelchair users, as one important part of the society who face miscellaneous difficulties on a daily basis; difficulties ranging from troubles in moving around and navigating in and outside home and accessing buildings, to secondary injuries that they usually undergo as a consequence of the above-normal and repetitive load they experience in their upper-body which they have to use for ambulation. We developed this study to simulate navigation using wheelchair in the VR, and since combination of navigation and VR has a high probability of inducing VIMS, we concentrated on mitigating VIMS while trying to ascertain higher sense of presence.

## This study

We have recently developed an immersive VR environment for manual wheelchair users that is comprised of a wheelchair ergometer in a VR cube (Fig 1). The virtual world is projected on the screens (three walls and the ground) and is controlled by the wheelchair user and not from outside: the interaction with the VR is through the natural and realistic means of pushing the wheels. Being able to see oneself when propelling the wheelchair adds to the immersion and is believed to intensify the feeling of presence. Presence is the perception of transportation to the virtual scene and the feeling of being there [16]. Presence is subjective and may vary from one

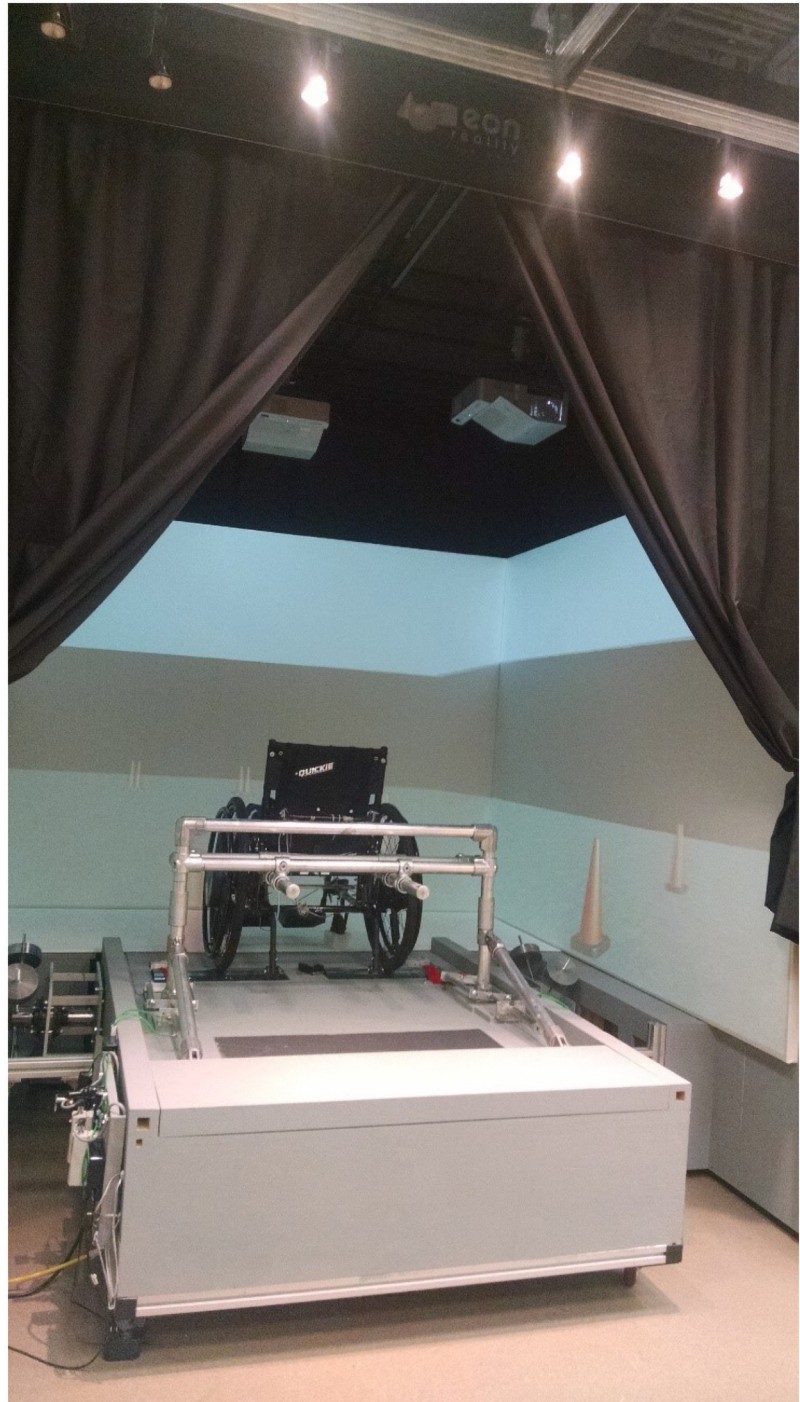

**Fig 1. Wheelchair ergometer inside EON IcubeTM Mobile.**

user to another in a given VR environment, while immersion is objective and deals with the level of sensory fidelity that is provided in a VR environment [16].

Three different approaches were taken to simulate wheelchair maneuvers in the VR environment, using three VR systems. To be brief, here we call the VR environment actuated using

each approach a "VR_sys". VR_sysI only mechanically replicates linear inertia (details on [17]), assuming rotational inertia can be neglected. VR_sysII and VR_sysIII take a mechanical (using pneumatic brakes to slow down rotations to induce the feeling of inertial resistance) and perceptual (slowing down rotations by a smart software application) approach in simulating rotational inertia, respectively, in addition to mechanically compensating linear inertia [17, 18]. Technical details regarding the development of the VR systems are provided in our former publication [17], as well as the literature review on the existing wheelchair simulators, so we did not repeat that here.

Having a wide field of view helps greatly in the feeling of presence [2] and immersion, which in turn helps VR users perform better in VR [4]; however, a wide field of view is shown to cause higher VIMS [2], as the peripheral vision which is responsible for detecting movements, is also more sensitive to fake movements (movements that are inaccurate, flickering, encompass lags, etc.) and can trigger motion sickness [19]. Immersion affects presence [4] and a VR with a holistic design would provide a higher sense of presence [20], but could this potentially increase the risk of VIMS?

The objective of this study was to assess motion sickness and virtual presence of participants when performing wheelchair maneuvers in each of those three systems, in addition to assessing the effect of up to four training sessions in acclimatizing to motion sickness. We hypothesized that up to four preconditioning sessions will suppress VIMS to an easily tolerable level, and also the VR systems tested in this study would receive a relatively high presence score.

This protocol was designed based on research studies that have reported that having participants trying VR in four [13], five [14, 21] and six [3] sessions had helped them acclimatized to motion sickness. These training sessions should be held on different days [8], as sleeping between the sessions helps to promote neuro-plasticity (repairing and forming new connections in the nervous system).

This study had an iterative approach to develop a VIMS-free VR with natural interaction for wheelchair users. We measured participants' VIMS and sense of presence in the simulator (VR systems) in 4–7 sessions. It is better said in advance that in our experiments a group of the same participants did not try different systems in equal time differences. Since we improved the VR environment based on the feedback received from the participants as the experiments progressed, the data obtained from each participant and for each session-system were different, however, we completed additional statistical analyses concerning the reduced sample size. The statistics were used with great care, but they were not the main outcome or focus of this work. Rather, this study was about the development of three system iterations in time, and how people reacted to training on one or many of these systems. However, we have made every attempt to draw statistically sound results from the experiments, based on standard statistical procedures. We also wish to note that although Statistics was not the focus of this research study, the statistical approaches were selected with care rather than to simplify the analyses. In order to get the most rigorous results, we carefully assessed each outcome for the six analyses to determine which of the parametric or nonparametric methods would suit the conditions of that dataset and chose the proper statistical approach accordingly. Hence, although sometimes not statistically significant, we believe that the results of this study are valuable and informative.

## Methods

### Participants

Fourteen healthy able-bodied (independent from using a wheelchair) subjects participated in this experiment. Small sample sizes around 10 are quite respectable when the research involves

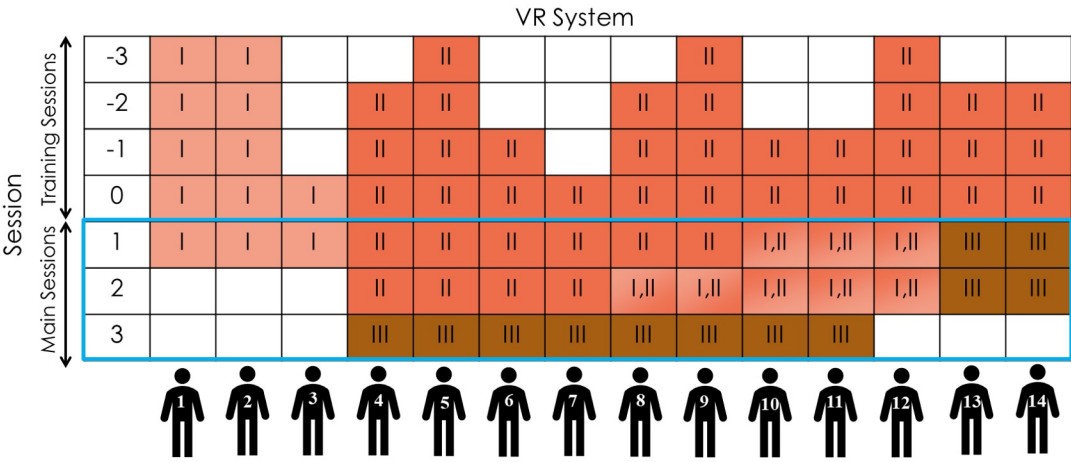

**Fig 2. Flowchart of the experiments.** This graph shows what systems each participant was exposed to, and in which sessions. Also, it shows how many training sessions each participant needed. Also, the blue rectangle marks the main sessions.

expensive, complex, and time-consuming study protocols. The experiment protocol was approved by the Human Research Ethics Board of the University of Alberta. Subjects -eight females and six males- signed a consent form and a ParQ & You physical readiness form prior to participating. They were 27.9 ± 4.74 years old on average, with no significant prior VR experience. Participants were asked in their first visit to hold a ruler on their forehead and look straight to the camera which was about 1 meter ahead of them, to record their intra-ocular distance. Participants tried different systems and had different numbers of training and main sessions, which is illustrated in Fig 2.

## Experimental procedure

On their first visit, participants were trained to propel a wheelchair in both VR and Real World (RW). Based on how they reacted in that session regarding motion sickness, some of them were asked to complete up to three more training sessions in preparation for participating in the main sessions. During the training sessions, participants were simply exposed to different VR scenes and asked to freely "move around" as long as they felt they liked it. The training sessions were finished, however, whenever the participant asked to stop. The duration of these sessions was between 5 to 30 minutes. All of the conditioning sessions were held on different days. The number of additional training sessions was determined based on the score of their motion sickness at the end of each training session and also by asking them directly if they feel ready to participate in the main experiments. The length of the sessions was up to the participants; they could stop the session as soon as they feel sick and bothered by the VR, or if they felt they have no problems (anymore) with being in the VR.

At the end of each training session, participants completed the Motion Sickness Assessment Questionnaire (MSAQ), a questionnaire that has been shown to be valid and reliable [22] in assessing different aspects of motion sickness. Subjects' readiness for participation in the main tests was determined based on their MSAQ score in the last training session and their written declaration that they feel ready and confident. Subjects then participated mostly in two main sessions, although some of the participants completed a third main session later on, as the progress of the VR systems demanded. Each of the main sessions consisted of performing a standard, reliable [23] and valid [24] agility test, Illinois agility test, with a wheelchair, both in RW and in the VR environment; they completed four trials of this test in VR and four trials in

RW. At the end of each main session, participants completed the MSAQ and also Igroup Presence Questionnaire (IPQ), a questionnaire used for assessing different aspects of feeling present in VR [25]. Most studies have tested the sense of presence in the VR using post-test questionnaires [26]. In this study, therefore, to measure the sense of presence in the VR systems we used the IPQ questionnaire, the validity and reliability of which has been shown using a large sample size (n = 296) [25].

## Materials

The VR environment consisted of a sophisticated wheelchair ergometer placed inside an EON IcubeTM Mobile (Fig 1): this VR environment allows a real-life-like experience of ambulation in VR by using the wheelchair as the interface to move around in VR. The wheelchair ergometer is equipped with an inertia system that replicates the biomechanics of straight-line wheelchair propulsion [17, 27]. The first VR system (VR_sysI) replicates straight-line biomechanically but does not provide inertial compensation for simulating turns. We hypothesize that when linear inertia is compensated, rotational inertia can be neglected and the participant's perception of turning can be induced using visual cues. In other words, the participant will use the wheels to turn as they do in RW, and the VR scene will follow the turn, but there will be no inertia resisting turning. Subjects recruitment started by using this system. It is worth stating that since not all the participants tried the same VR systems in their experiments, the data of each participant was included in the analysis of the system(s) that they had tried.

After recruiting a few subjects (Figs 2 and 3), screening the comments they made during their visit led us to believe that lacking the inertial compensation for wheelchair turning triggers motion sickness and is hard to tolerate and get used to, and also, it did not "feel like the real world". Therefore, VR_sysII was designed and built: a system that uses a pneumatic braking system to simulate rotational inertia by adding friction to one of the wheels when turning,

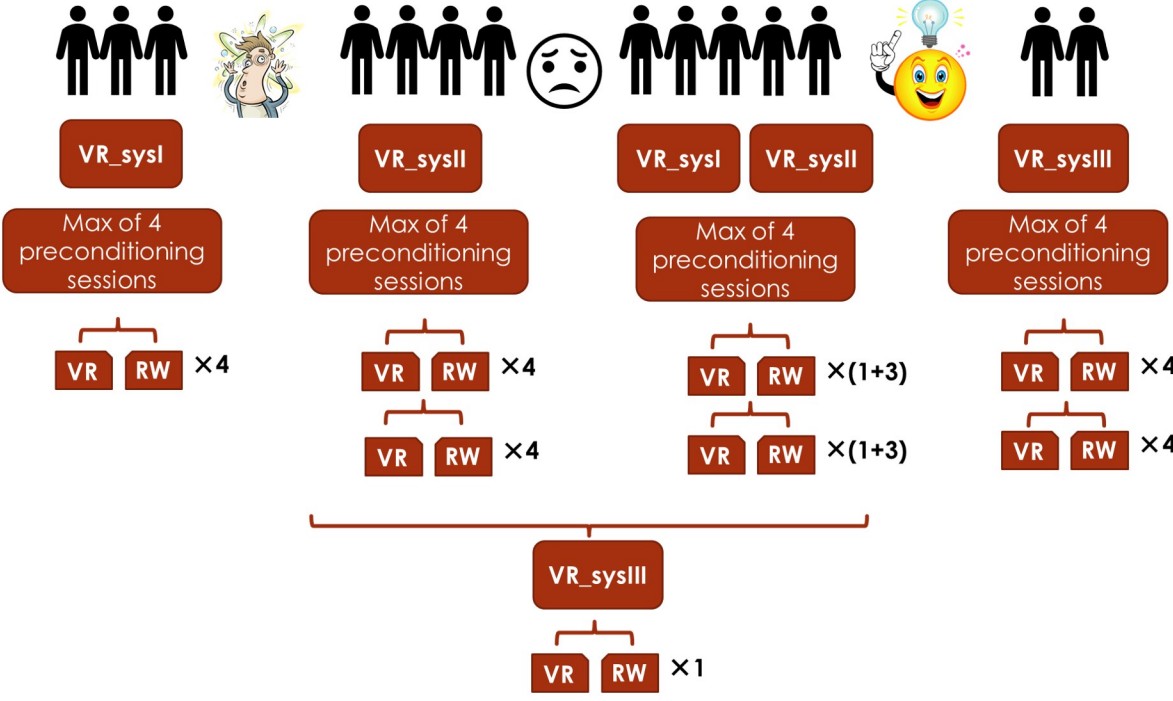

**Fig 3. Summary of the experiments.** Experiments started on the left side and proceeded to the right side.

as needed. However, as this system uses a pneumatic system, alternating between the setpoint braking pressures builds up hysteresis, making it impossible to formulate and control the set pressures. To solve this problem, the pneumatic pressure was reset back to 0 between every two set points; although this was an effective solution for setting the exact pressure needed for rotational inertia replication, inevitably the limited response time in the system could not be better than ~10 Hz. When the participant wanted to turn fast and pushed harder on one of the wheels, sometimes they could not see the corresponding turn in the VR display. As a result, they pushed harder and tended to overshoot. This made the control of the VR_sysII to be rather hard.

Participant recruitment was resumed using VR_sysII; After recruiting a few more participants and based on the feedback received from them, it was realized that although helpful at the beginning, participants started being dissatisfied with VR_sysII after acclimatizing to VR. This was because VR_sysII was harder to control, due to the slow response time of the pistons. Thus, to compare the two systems, one VR trial and one RW trial of each session were completed using VR_sysI and the rest of the trials of that session were completed using VR_sysII (see Figs 2 and 3).

A third system, VR_sysIII, was developed to deal with the limitations of VR_sysII: Instead of mechanically compensating the rotational inertia, this system induced the perception of rotational inertia by slowing down the rotation in the visual feedback of the VR display, thereby increasing the effort needed to rotate the scene. Details of the VR systems and their validity and reliability can be found in our former publications [17, 18]. When this system was ready, it was tested by two participants in their main sessions. Their feedback showed greatly improved satisfaction. Therefore, former participants were called back to try VR_sysIII in an additional main session. Figs 2 and 3 show a summary of the experiments. In Fig 2, the first column shows the session numbers (-3 to 0 indicate preconditioning sessions and 1 to 3 indicate the main sessions), and the other columns are indicative of the VR systems tried by the participant. In addition to VR, participants also performed real-world tests in all main sessions.

Here is why the sessions are named -3 to 3: Participants took 1 to 4 training sessions based on their needs. Since the number of the training sessions was different for different subjects and since the training sessions preceded the main sessions, the last training session was named 0 and the other training sessions were named retrospectively. This way, the first main session which came after the last training session is named session 1 for all participants, while keeping the chronological sequence of all sessions for everyone.

Table 1 shows the technical descriptions of the VR systems that relate to VIMS.

The data then were classified into 7 sessions: from session -3 to session 3. Session -3 to 0 being training sessions, and session 1 to 3 being the main sessions. This was designed based on the fact that participants had different counts of training sessions from one to four; so, the sessions were named by how far they were from the first main session. This way all main sessions start at 1 while keeping the sequential order of sessions. This is also rationally the right choice, as people who needed fewer training sessions were those who were less susceptible to motion sickness and so generally scored less for MSAQ in their first training session.

The MSAQ [22] consists of 16 questions in four subcategories (5 scores): total (T), gastrointestinal (GI), central (C), peripheral (P), and sopite-related (S). In the original MSAQ each question has a score from 1 to 9; so, the total score is the sum of all scores that will be a number between 11 and 144. The four subcategories also follow a similar rule. This scale was not intuitive in the context of this study, so we slightly modified the MSAQ to make the scores more understandable: we changed the scale of each question to 0 to 9 and at the end, scaled the total score and all subcategory scores to a maximum of 100. This way, all scores are from 0 to 100,

**Table 1. Technical descriptions of the VR systems.**

| Factor | Our VR systems | Known thresholds |
|---|---|---|
| Time lag | <10 ms | <10 ms [6] |
| Frame rate | 60 Hz | >10 Hz [10] |
| Haptics | Inertia is felt immediately by hand when pushing the wheels, as the systems are not motorized | - |
| Response time or interactivity (for VR_sysII) | Up to 0.1 s | <0.1 s* [10] |
| Control system | Proportional | - |
| Inter ocular distance | Measured for each participant and accordingly adjusted in the simulation | - |

*- May not be enough if there are fast-moving objects in the scene that are not controlled by the user [10].

which helps with making judgments and comparisons much easier. For the training sessions, one question was added at the end of MSAQ to directly ask if they felt ready to participate in the main experiments.

The IPQ contains four subcategories, hidden in 14 questions: INVolvement (INV), Experienced Realism (ER), Spatial Presence (SP), and General (G). In addition to the original 14 questions, we added one more question to the IPQ to specifically ask them about their idea of similarity/difference between RW and VR, as in Fig 4.

## Statistical procedure

Small sample sizes reduce the power of studies. On the other hand, the parametric methods are more powerful than non-parametric methods. Since the power of this study for our research questions (that involve high between-subject variations) is fragile, we used every effort to add to the power of the analyses. Thus, where possible, parametric methods were used. However, if the statistical assumptions were not met, the non-parametric methods were employed. In other words, we used parametric and nonparametric tests when appropriate. SPSS (IBM® SPSS® Statistics Premium GradPack 24 for Windows) was used for the statistical analyses.

For each research question, data were checked for normality first, using Shapiro-Wilk test which is highly recommended for testing normal distribution in SPSS [28]; If not normal, an attempt was made to make data follow normal distribution using the two-step method [29] (or other transformation functions). Then, for any subcategories that followed a normal distribution, MANOVA was used to test the null hypothesis. In the case of non-normally distributed subcategories, we used non-parametric methods to test the null hypotheses. Data for groups involved were checked firstly for having similar distributions, and secondly for satisfying the Assumption of Homogeneity of Variances (AHV). If assumptions were met, Kruskal-Wallis method was used to find whether there were any statistically significant results.

How much similar were the forces you needed to apply to turn a given angle, i.e., 45 degrees?

VR needed much less force — the same — VR needed much more force
0 —————————————— 5 —————————————— 10

**Fig 4. The questions added to IPO.**

**Table 2. IPQ data available for each participant.**

| Subject | 1 | 2 | 3 | 4 | 5 | 6 | 7 | 8 | 9 | 10 | 11 | 12 | 13 | 14 |
|---|---|---|---|---|---|---|---|---|---|---|---|---|---|---|
| VR_sysI | 1 | 1 | 1 | | | | 2 | 2 | 2 | 1,2 | 1,2 | 1,2 | | |
| VR_sysII | | | | 1,2 | 1,2 | 1 | 1,2 | 1,2 | 1,2 | 1,2 | 1,2 | 1 | | |
| VR_sysIII | | | | 3 | 3 | 3 | 3 | 3 | 3 | 3 | 3 | | 1,2 | 1,2 |

Numbers before each VR system indicate the session numbers that participants filled in the IPQ.

**Table 3. MSAQ data available for each participant.**

| Subject | 1 | 2 | 3 | 4 | 5 | 6 | 7 | 8 | 9 | 10 | 11 | 12 | 13 | 14 |
|---|---|---|---|---|---|---|---|---|---|---|---|---|---|---|
| VR_sysI | -1,0,1 | -1,0,1 | 0,1 | | | | | 2 | -3,2 | 1,2 | 1,2 | 1,2 | -1 | 0 |
| VR_sysII | | | | -2,-1,0, 1,2 | -3,-2,-1,0, 1,2 | -1,0, 1,2 | 0,1,2 | -2,-1,0, 1,2 | -2,-1,0, 1,2 | -1,0, 1,2 | -1,0, 1,2 | -3,-2,-1,0, 1,2 | -2,0 | -2,-1 |
| VR_sysIII | | | | 3 | 3 | 3 | 3 | 3 | 3 | 3 | 3 | | 1,2 | 1,2 |

Numbers before each VR system indicate the session numbers that participants filled in the MSAQ.

Repeated measures MANOVA was not used here as the data in this study could not be paired between groups, e. g. a group of the same people did not try different systems in equal time differences. Since we improved the VR environment based on the feedback received from the participants as the experiments progressed, the data obtained from each participant and for each session-system were different. Tables 2 and 3 show data available for each participant.

## Results

Our research questions are addressed here, by analyzing the data obtained from this study. Tables 2 and 3 were used to extract the statistically validated data for testing each analysis. Details of and the grounds for opting the statistical procedures on the reported results of this study are shown in Tables 4–6.

### MSAQ

**Training sessions.** Participants took one to four training sessions based on their needs until they were ready to start the main sessions. The average time gap between the training sessions was 8.1 days (SD = 7.78 days). Table 7 shows the numbers of participants needing different numbers of training sessions. The mean number of sessions needed to mitigate VIMS was 2.8, with a standard deviation of 1.1.

*Analysis 1*: *The influence of training sessions in decreasing motion sickness*. MANOVA was significant with a total observed power of 0.91 and a total effect size of 0.72. A post hoc analysis revealed that the subcategories of T, GI, and C showed significant results; therefore, the training sessions significantly helped participants by decreasing those three factors of motion sickness (total, gastrointestinal, and central. Fig 5 shows how the average MSAQ score of all data (14 subjects) diminished from the first training session to the last one, showing that the training sessions indeed work. Also, all participants replied "yes" to the direct question they were asked that whether they felt ready to take part in the relatively long sessions of a complex maneuvering task, after a maximum of four sessions.

**Table 4. Details of statistical procedures for the analyses 1 to 5.**

| Analysis | Data | MSAQ or IPQ? | Categories not initially having normal distribution | Normal after transformation? | Which transformation method? | AHV is met? | Pillai's Trace significance of MANOVA | F value |
|---|---|---|---|---|---|---|---|---|
| 1: The influence of training sessions in decreasing motion sickness | First training sessions' MSAQ_sysII (Sample Size (SS) = 9) versus last training sessions' MSAQ_sysII (SS = 10). | MSAQ | GI and P | Yes | 2-step method | Yes (Box's M test significance of 0.035 [30]) | 0.008 | 5.573 |
| 2: VR_sysII vs. VR_sysI during the training stage. | MSAQ scores of the training sessions: VR_sysI (SS = 6) versus VR_sysII (SS = 11). Data were averaged where there was more than one data per system per participant | MSAQ | None | – | – | Yes (Box's M test significance of 0.994) | 0.285 | 1.438 |
| 3: Comparing the VR systems during the main sessions, for MSAQ scores. | MSAQ scores of the main sessions: VR_sysI (SS = 8) versus VR_sysII (SS = 9) versus VR_sysIII (SS = 10). Data were averaged where there was more than one data per system per participant. | MSAQ | G, C, and P (at least for one system) | Only for C- See Table 11 for non-normal variables (G and P) | 2-step method | Yes (Box's M test significance of 0.076) | 0.414 | 1.038 |
| 4: Comparing VR systems regarding the IPQ scores | IPQ scores of the main sessions: VR_sysI (SS = 9) versus VR_sysII (SS = 9) versus VR_sysIII (SS = 10). Data were averaged where there was more than one data per system per participant. | IPQ | G, SP | No-See Table 11 for these variables (G and SP) | – | Yes (Box's M test significance of 0.734) | 0.128 | 1.886 |
| 5: Comparing the IPQ scores among sessions | IPQ scores of the main sessions wherever IPQ of at least 2 sessions are available: Session 1 (SS = 11) versus session 2 (SS = 9) versus session 3 (SS = 8). Data were averaged where there was more than one system tried per session per participant. | IPQ | None | – | – | Yes (Box's M test significance of 0.828) | 0.066 | 2.014 |

Fig 6 depicts the mean and SD of all subcategories of MSAQ for the first and the last training sessions. MSAQ scores are from 0 to 100.

*Analysis 2: VR_sysII vs. VR_sysI during the training stage.* All subcategories (T, GI, C, P, and S) had a normal distribution for each group (mean and standard deviations are presented in Table 8). MANOVA results were not significant and the mean data from Table 8 also do not show any clinical impact. Therefore, there was no statistically meaningful difference

**Table 5. Details of statistical procedures for the analysis 6.**

| Analysis | Data | Assumptions |
|---|---|---|
| 6: Correlation of IPQ and MSAQ scores | All IPQ and MSAQ (SS = 14) scores of all sessions, averaged for each participant, so we are left with only one column of IPQ and one column of MSAQ (one correlation is assessed). According to the former results of analyses, it is permittable to average for each participant. | - Normal distribution: P-value of Shapiro-Wilk test of normality for total motion sickness = 0.279 and for general presence = 0.066 = > *Assumption met*—Linearity: P-value of deviation from linearity: 0.93 = > *Assumption met* |

**Table 6. Details of statistical procedures for the non-parametric tests.**

| Analysis | Non-normally distributed categories | Statistical test | AHV met?* |
|---|---|---|---|
| 3 | G, P | Kruskal-Wallis | Yes- Significance = 0.998 |
| 4 | G, SP | Kruskal-Wallis | Yes- Significance = 0.43 |

*- To test the underlying assumption of Kruskal-Wallis,
AHV, we ranked all data of the three systems, then found the average of the ranked data for each system and found the absolute difference of each ranked data from the mean of the respected group, and finally, found out if there were real differences between those absolute difference data, using MANOVA. This is the non-parametric equivalent of Levene's test.

**Table 7. Number of participants (#females, #males) taking each number of training sessions.**

| # Training sessions | 1 | 2 | 3 | 4 |
|---|---|---|---|---|
| How many participants? | 2 (2f) | 3 (2f, 1m) | 4 (2f, 2m) | 5 (2f, 3m) |

between the VR systems I and II during the training sessions in terms of the measured motion sickness.

**Main sessions.** *Analysis 3*: *Compering the VR systems during the main sessions, for MSAQ scores*. Kruskal-Wallis test was used for P and G (see Tables 4 and 6) that returned an exact significance of 0.388. Therefore, there is no meaningful difference between different VR systems during the main experiments (stabilized motion sickness).

For subcategories total, central, and sopite-related, MANOVA test was not significant with the observed power of only 36.5%main; mean and SD of these subcategories is depicted in Fig 7.

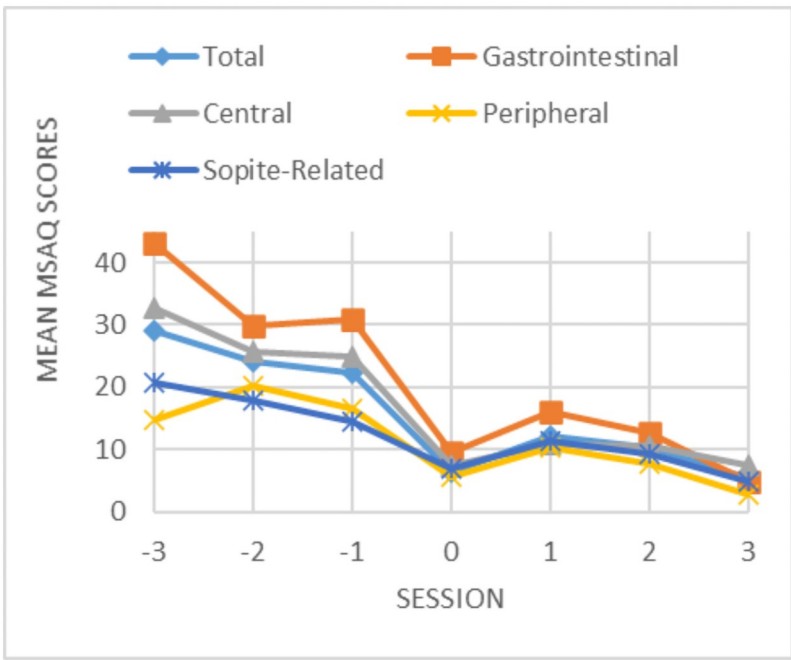

**Fig 5. Average MSAQ score of all data for each subcategory.**

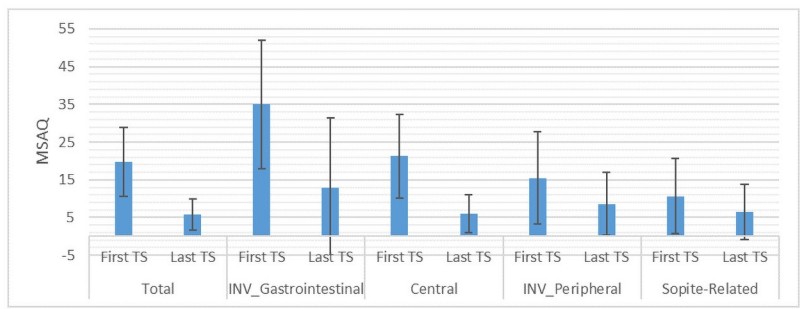

**Fig 6. Mean and SD of subcategories of MSAQ for the first and the last training sessions.** The prefix INV- indicates outcomes that needed transformation to comply with the normal distribution condition.

## IPQ

**Analysis. 4: Comparing VR systems regarding the IPQ scores**. We used a non-parametric method (Kruskal-Wallis) for analyzing the results of G and SP, and for the others (INV and ER) we used MANOVA which is more powerful (details on Tables 4 and 6).

*Kruskal-Wallis*. The P-value of 0.19 and 0.23 for Kruskal-Wallis test for G and SP suggested that there is no meaningful difference between the VR systems with regard to G and SP. Also,

**Table 8. Mean and SD of MSAQ subcategories for the training sessions.**

| MSAQ subcategory | Total | | Gastrointestinal | | Central | | Peripheral | | Sopite-related | |
|---|---|---|---|---|---|---|---|---|---|---|
| VR system | I | II | I | II | I | II | I | II | I | II |
| Mean | 24.9 | 23.4 | 31.0 | 32.6 | 32.6 | 26.5 | 19.8 | 16.2 | 13.0 | 15.9 |
| Standard Deviation | 18.3 | 16.0 | 28.9 | 24.1 | 26.4 | 19.1 | 23.0 | 14.0 | 10.5 | 16.2 |

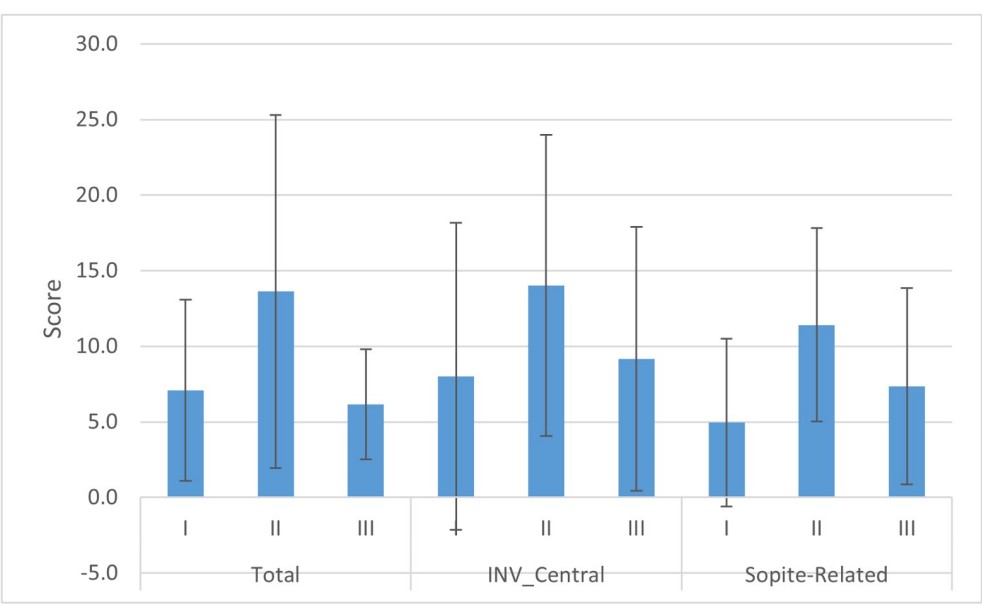

**Fig 7. Mean and SD of MSAQ subcategories for the main sessions.** The prefix INV- indicates that transformation was needed to comply with the normal distribution condition.

**Table 9. Median of general and spatial presence (Scores are out of 6).**

| VR system | Median | | | |
|---|---|---|---|---|
| | I | II | III | Total |
| **Spatial Presence** | 4.4 | 4.2 | 5.1 | 4.4 |
| **General** | 4.3 | 4 | 5 | 4.5 |

the effect sizes of 0.13 and 0.11 for G showed that about 10 percent of the variability in rank scores of G and SP are accounted for by the VR systems.

The median of G and SP scores for each VR system and all VR systems as a whole are presented in Table 9.

*MANOVA.* MANOVA showed that the three systems were not statistically different. Table 10 presents the mean and standard deviation of normally distributed presence factors for each system and all data.

Although no meaningful differences were detected between the VR systems regarding the presence factors, looking at means (Fig 8) we could see an almost linear trend from VR_sysII receiving the lowest scores to VR_sysIII receiving the highest scores for involvement and experienced realism. Therefore, *linear contrast analysis* was performed on INV and ER that revealed a significant linear trend from VR_sysII to VR_sysI to VR_sysIII (sig = 0.033 and 0.012, respectively) with good observed-powers (0.58 and 0.74) and considerable effect sizes (0.17 and 0.23). In other words, about 20% of the variability in INV and ER scores is accounted for by the VR system.

**Analysis 5: Comparing the IPQ scores among sessions**. MANOVA results showed no significant difference among sessions for IPQ, with a significance of 0.066 and observed power of 75.4%.

**Analysis 6: Correlation of IPQ and MSAQ scores**. Pearson coefficient of correlation was obtained as -0.533 which is significant for a sample size of 14. This is an interesting result since it shows the more the person suffers from motion sickness, the less they grade their presence in VR, confirming an interaction between VIMS and "presence".

## Direct question

Question number 15 (Q15) had a normal distribution. ANOVA was used to find possible statistical differences between the scores given to each system for Q15, which was rejected with 23.5% observed power. Table 11 shows the descriptive statistics for each system and Fig 9 illustrates how far from similar are the mean scores of each VR system. According to these results, with system III, participants generated the forces that are the closest to the RW conditions.

## Free text

At the end of each session, participants were asked to write down any comments that they had about their experience. Word clouds, which are a common way to represent qualitative data [31–33], were then made from the comments gathered for each system and are shown in

**Table 10. Mean and SD of presence factors (out of 6).**

| VR system | Mean (SD) | | | |
|---|---|---|---|---|
| | I | II | III | Total |
| *Involvement* | 4.2 (0.9) | 3.7 (0.8) | 4.5 (0.6) | 4.1 (0.8) |
| *Experienced Realism* | 3.5 (1.2) | 2.7 (0.9) | 3.9 (0.7) | 3.4 (1) |

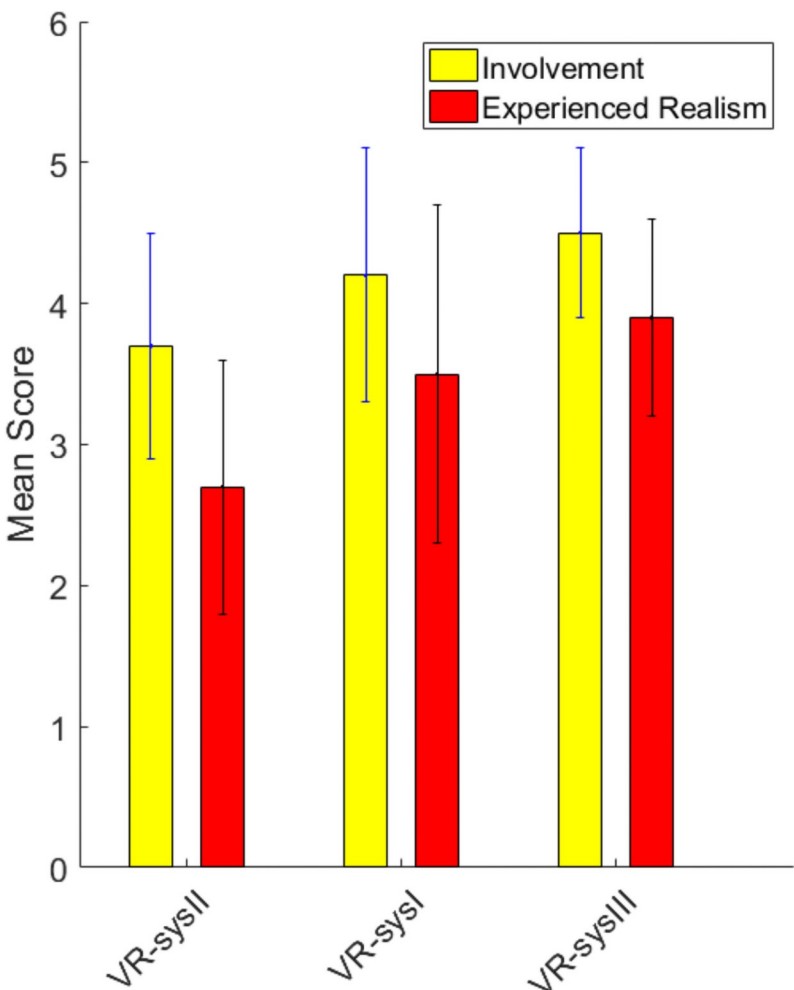

**Fig 8. The linear trend for INV and ER among the three systems.** The system with the lowest scores (VR_sysII) was placed far left and the system with the highest scores (VR_sysIII) was placed far right on the horizontal axis.

**Table 11. Descriptive statistics for Q15.**

|  | Mean | SD |
|---|---|---|
| VR_sysI | 3.6 | 2.6 |
| VR_sysII | 6.5 | 4.4 |
| VR_sysIII | 5.4 | 1.3 |

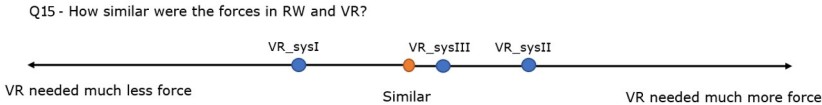

**Fig 9. Distances each system score has from the ideal situation (absolute similarity).**

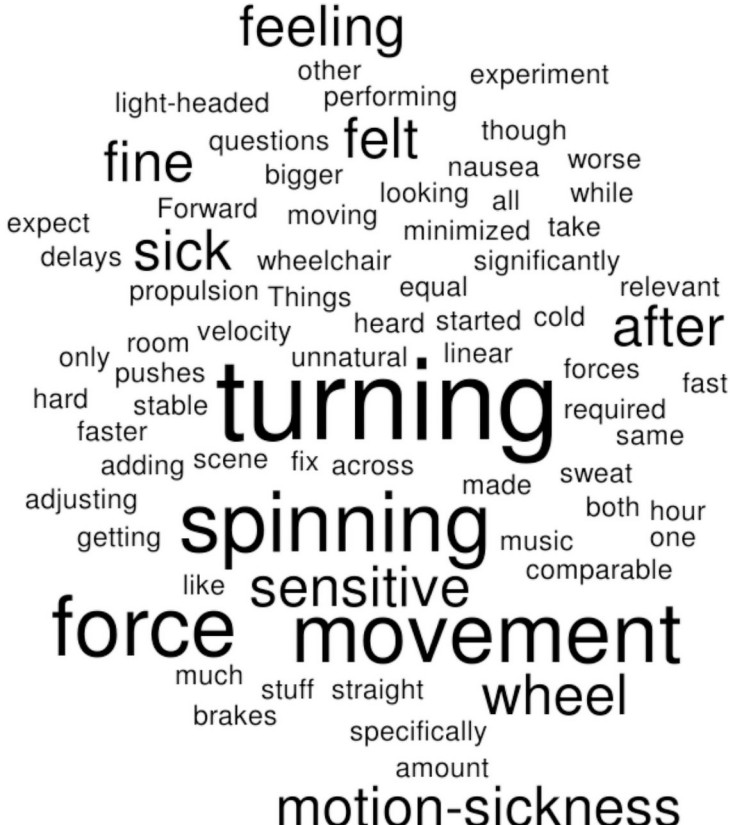

**Fig 10. Word cloud for VR_sysI (made using [34]).**

Figs 10–12. In these figures, the size of each word is related to the rank of its repetition in the participants' comments. Thus bigger words indirectly show that those words are concerns to the participants. Note that for each word cloud, frequent but non-informative words were excluded, namely: VR, RW, real, virtual, world, some, more, and time.

As the word clouds clearly show, participants were mainly displeased with their experience of VR_sysI, complaining about it being too sensitive to turning, so much so it would make them sick and feel like "the world is spinning". For VR_sysII, however, they did not talk about sensitivity and spinning anymore, but they were not satisfied with the system yet, either. Scenes were sometimes slower and not how they would expect for turning. For the VR_sysIII though, they generally wrote about how this time it was better, smoother, the easiest for their motion sickness, and more realistic. These comments nicely show how the VR systems gradually improved based on the participants' feedback.

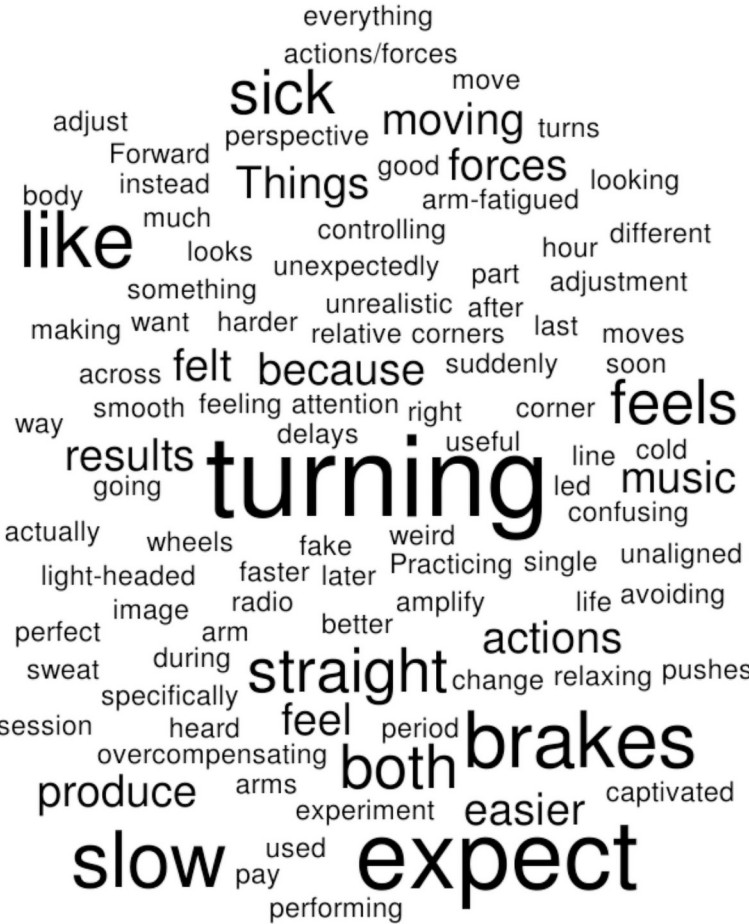

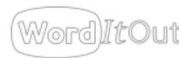

**Fig 11. Word cloud for VR_sysII (made using [34]).**

## Discussion

In this study, motion sickness and sense of presence in three VR systems developed for wheelchair maneuvering were compared. Also, the effect of providing up to four training sessions in reducing motion sickness to VR to a tolerable level was confirmed. Since the discussion on IPQ and MSAQ results are intertwined, we first discuss MSAQ scores alone and then discuss the general results of this study, considering both IPQ and MSAQ results.

### MSAQ

**Training sessions.** According to the results of this study, the training sessions significantly reduced the gastrointestinal and central motion sickness, as well as the total motion sickness level, with a very high effect size (0.72). Also, the most important evidence and the one

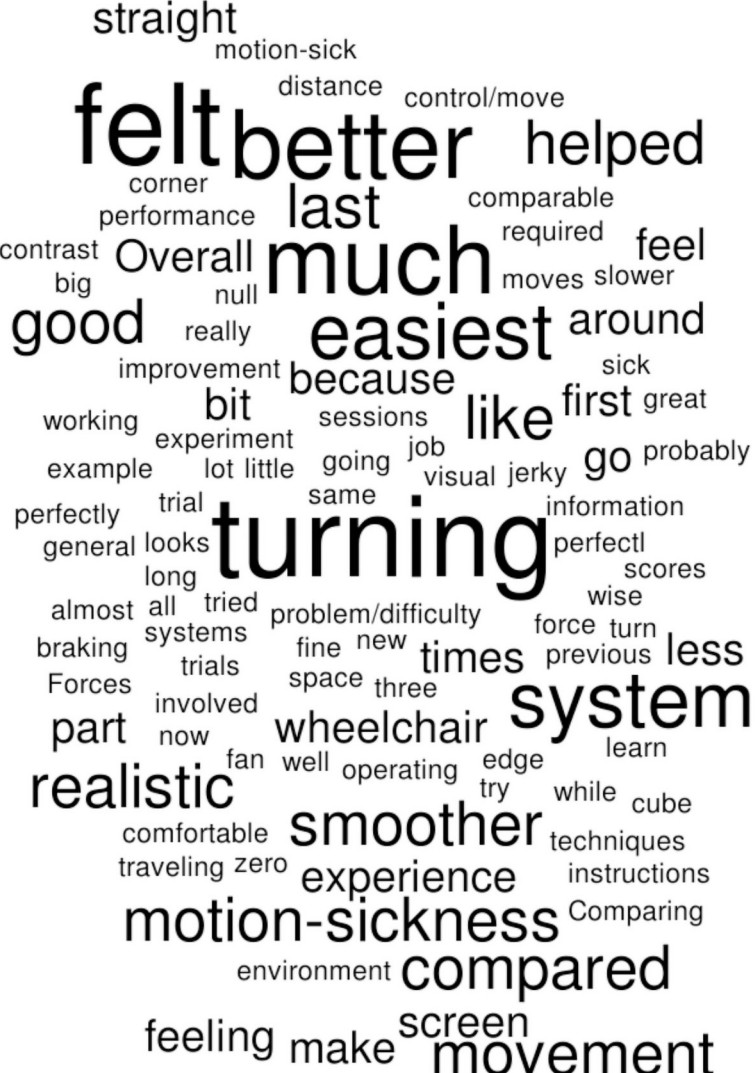

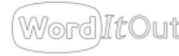

**Fig 12. Word cloud for VR_sysIII (made using [34]).**

conclusive outcome for the effect of the training sessions in reducing VIMS to a minimal/tolerable level is the question they were asked whether they felt ready to take part in the relatively long sessions of a complex maneuvering task to which all participants replied "yes" after a maximum of four sessions. This clearly shows that the maximum of four training sessions does indeed work in extinguishing motion sickness.

This is a very positive and encouraging result, as one of the main issues of usability of VR is still the motion sickness that it causes in many of its users [1, 19, 35]. Motion sickness, in addition to all the negative consequences and unfavorable feelings accompanying it, could be unsafe, as was experienced by one of the participants of this study. When he tried VR for the first time, he suddenly became motion sick. He was helped out of the VR and recovered.

However, he reported later that when he went home on his bike, he still was disoriented and had vertigo, and did not feel safe riding. This clearly illustrates the importance of looking after, predicting, and controlling VIMS for the safety of VR users.

Although peripheral and sopite-related categories of MSAQ did not show statistically significant results, the downward trend in these motion sickness categories, as it is depicted in Fig 6, suggests potential clinical relevance. The observed power for T, GI, C, P, and S was, respectively: 0.98, 0.64, 0.95, 0.25, and 0.16, which shows a good to excellent power for the first categories and albeit a great probability of type II error (false negative) for P and S (β = 75% *and* 84%). This is a consequence of the high standard deviation of the data, especially for P and S, which is a result of the big between-subject variations for motion sickness (see the overlapping standard devotions).

**Main sessions.** For subcategories total, central, and sopite-related the difference between the systems during the main sessions was not statistically meaningful. However, looking at the mean data (Fig 7) we see a considerable difference between MSAQ scores of VR_sysII and the other systems which is indicative of potential clinical impact, but more research is needed to conclude this. Here again, the large between-subject variability in the propensity for motion sickness has caused high standard deviations and high chances of type II error.

## General discussion on IPQ and MSAQ results

Using questionnaires is a good way to measure presence as it is cheap and easy, does not interrupt the experiment, and has high face validity [26]. However, the recency effect is one of the main disadvantages of questionnaires [26] which means the scores participants give to the questions about presence are usually the way they were feeling at the final parts of the test. On the other hand, motion sickness is a cumulative construct and it builds up as time passes during the experiment [5] and gradually tends to throw the participant's concentration away [8, 36]. This means that participants who, after receiving some training, still have some susceptibility to motion sickness have probably given scores that show more sickness than how they have felt during the whole experiment.

Despite all that, in general, the three VR systems studied here received relatively low motion sickness and high virtual presence scores during the main sessions, which is indicative of their good general quality.

It was shown in this paper that direct questions that were added to the MSAQ and IPQ questionnaires provided stronger evidence about user preference regarding sense of presence and also the overall feeling of motion sickness. Although great care was taken when selecting validated questionnaires to characterize motion sickness and virtual presence well, this seemed to be inadequate to capture users' ideas about our VR systems. Therefore, it was felt that there was a need for some objective questions. This indicates a serious shortcoming in evaluating VR using currently available validated tools.

With regards to the difference among the VR systems, no statistically meaningful difference was detected for either MSAQ or IPQ. It should be said that the focus of this research was on iterative wheelchair virtual reality development with the users' comments being considered during this continuous development. This approach to development allows fast consideration of user's perception and faster iterations between VR systems; however, it does introduce complexity in analyzing data post-experiment. As a result of high between-subject variances, which are due to natural differences between people rather than a consequence of the study methods, much bigger sample sizes are needed for significant results. However, future work should focus on creating a VR user experience that is a significant improvement on the designs used

in this study, rather than recruiting a larger sample of participants, simply to demonstrate the differences between these systems.

Based on the participants' comments (word clouds), the VR_sysIII was the easiest motion-sickness-wise and the most realistic one. Questionnaires' results also, although not statistically significant, show higher IPQ scores for VR_sysIII and lower IPQ scores for VR_sysII. This indicates that users were more inclined to choose and be satisfied with VR_sysIII and the least so with VR_sysII.

Interactivity constraint is an important factor that influences the sense of presence in VR [10]. It is the time it takes from when the participant provides an input to the VR system to when he/she feels/observes the effect of it; in our case, since the participant applies a force to the wheels to when he/she observes the corresponding displacement/rotation in the visual feedback. Interactivity constraint was a considerable issue in the VR_sysII, as it was based on a mechanical system that encompasses some delay in its application.

The statistics are not the main outcome of this work, but rather, this study is about the development of three system iterations in time, and how people reacted to training on one or many of these systems. As already mentioned, the statistical analyses failed to detect any differences among the systems, either for IPQ or MSAQ, although participants' comments caused us to anticipate some. Also, as mentioned earlier, two of the five subcategories of motion sickness (SR and P) were not shown to significantly decrease from the first training session to the last one, despite the large decrease in their score (percent of reduction for each subcategory were: T: 70.6, G: 63.1, C: 71.8, P: 44.5, and SR: 40.2). The large standard deviations in most data in this study led to relatively low study power and therefore a high chance of type II error, and thus, some of the differences detected in this study could not be confirmed statistically. Despite this, we believe that the results of this study have important clinical and practical values.

The sample size of 10 that was used for testing most of the hypotheses here, is quite respectable when the research involves expensive, complex, and time-consuming study protocols. Nevertheless, large between-subject and possibly small effect size variations indicate that larger sample sizes are needed to detect meaningful effects from natural variations. A retrospective power analysis based on the effect sizes obtained in this study, for example, shows that for testing the effects of training sessions in eliminating peripheral and sopite-related motion sickness (Analysis 5), we need to recruit 628 and 1751 subjects, respectively [37] (power = 0.8 and $\alpha$ = 0.05)! The same observation was made by another study on simulator sickness that was unable to find significant results and stated that a much bigger sample size is needed due to great inter-individual differences of the study participants [8].

The third session took place about four/five months after the second session and we believe this has biased the IPQ scores of the third session, as the VR was not so exciting and a new experience anymore and thus received lower scores.

According to the results of this study, there is a meaningful inverse relationship between the level of motion sickness in the VR and the level of presence the VR users experience, which is consistent with similar results of other studies [35, 38]. In other words, for the VR users to have a realistic experience, it is important to make sure that the VR is carefully designed and calibrated to minimize issues that throw the users off and trigger motion sickness. Additionally, it is necessary to ensure that the users take enough training sessions to eliminate/minimize nausea and therefore, increase the usability of the VR by enhancing the realism of the experience.

## Conclusion

The motion sickness and sense of presence of the participants in three VR systems were assessed in this study. The effect of providing up to four training sessions to precondition

participants to VR was also assessed. We found that the training sessions significantly reduced the gastrointestinal and central motion sickness, as well as the total motion sickness level, with a very high effect size (0.72). This is a very positive and encouraging result, as one of the main issues for the usability of VR is that motion sickness causes many [15, 35, 39] to abandon VR, or at least feel uncomfortable using it. In general, the three VR systems studied here resulted in relatively low motion sickness and high virtual presence scores during the main sessions, which is indicative of their good general technical and experimental quality. With regards to the difference among the VR systems, no statistically meaningful difference was detected for either MSAQ or IPQ. Nevertheless, based on the participants' comments, VR_sysIII was the most comfortably tolerated and the most realistic one. Questionnaires' results showed that although not statistically significant, there were higher IPQ scores for VR_sysIII and lower IPQ scores for VR_sysII. Thus, we can conclude that based on surveys and qualitative data, VR_sysIII and VR_sysII gained the most and the least user preference, respectively.

One key observation in this study was that when simulating wheelchair manoeuvres, the technical and perceptual challenges of simulating turning are the main issue, especially when the rotational inertia is totally absent (VR_sysI). For having both a biomechanically sound wheelchair simulation and user satisfaction, it is important to compensate for both linear and rotational inertia while providing smooth visual feedback; something that complies with user expectations.

## Acknowledgments

The authors would like to thank the participants of this study for their time and great comments that helped to refine our VR systems.

## Author Contributions

**Conceptualization:** Zohreh Salimi, Martin William Ferguson-Pell.

**Data curation:** Zohreh Salimi, Martin William Ferguson-Pell.

**Formal analysis:** Zohreh Salimi.

**Funding acquisition:** Martin William Ferguson-Pell.

**Investigation:** Zohreh Salimi.

**Methodology:** Zohreh Salimi, Martin William Ferguson-Pell.

**Project administration:** Zohreh Salimi.

**Resources:** Zohreh Salimi, Martin William Ferguson-Pell.

**Software:** Zohreh Salimi.

**Supervision:** Martin William Ferguson-Pell.

**Validation:** Zohreh Salimi, Martin William Ferguson-Pell.

**Visualization:** Zohreh Salimi.

**Writing – original draft:** Zohreh Salimi.

**Writing – review & editing:** Martin William Ferguson-Pell.

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
