## [Decision Letter · Decision Letter 0]

26 Feb 2021

PONE-D-21-03454

Motion sickness and sense of presence in three virtual reality environments developed for manual wheelchair users

PLOS ONE

Dear Dr. Salimi,

Thank you for submitting your manuscript to PLOS ONE. After careful consideration, we feel that it has merit but does not fully meet PLOS ONE’s publication criteria as it currently stands. Therefore, we invite you to submit a revised version of the manuscript that addresses the points raised during the review process.

I have secured reviews from three subject-matter experts. As you will see, each identifies significant problems with the submitted manuscript. Please carefully revise, based on these detailed comments.

We look forward to receiving your revised manuscript.

Kind regards,

Thomas A Stoffregen, PhD

Academic Editor

PLOS ONE

Journal Requirements:

Reviewers' comments:

Reviewer's Responses to Questions

**Comments to the Author**

1. Is the manuscript technically sound, and do the data support the conclusions?

Reviewer #1: Yes

Reviewer #2: No

Reviewer #3: Yes

2. Has the statistical analysis been performed appropriately and rigorously? 

Reviewer #1: Yes

Reviewer #2: No

Reviewer #3: No

3. Have the authors made all data underlying the findings in their manuscript fully available?

Reviewer #1: Yes

Reviewer #2: Yes

Reviewer #3: Yes

4. Is the manuscript presented in an intelligible fashion and written in standard English?

Reviewer #1: Yes

Reviewer #2: Yes

Reviewer #3: Yes

5. Review Comments to the Author

Reviewer #1: Line 34: I'm confused. VR system or environments? System reads as the physical hardware, the actual system that the VR environment is created by.

Line 18: The users' sense of presence is a major focus of this paper. Defining exactly what that is would strengthen your paper immensely. Is this the sense of immersion? (I see in line 80 that these are listed as two separate things.) Please define this term.

Line 54: Missing period after [8].

Line 58: It is "stimuli."

Line 65: VR environment again, verses system. This is later changed in the Materials section (line 134). If these two are the same thing, it would be best to chose a single term. If there is some major difference between "environment" vs "system," that terminology should be established in the Introduction section.

Line 89: Why four? I would greatly appreciate an explanation in this paragraph about why this number was chosen, as well as any citations that may support this.

Line 89: Numbers under 10 (that are not reference numbers) are spelled out.

Line 102: What was the exclusion criteria? Motion sickness can also be affected by a history of concussion, TBI, etc. What does "able-bodied" actually mean?

Line 109: Again, with spelling out numbers under 10 that are not references. Make sure this is consistent throughout the manuscript.

Line 110: I did not initially realize that the sessions were held on different days. This should be clarified sooner, as the past tense used in line 117 confused me to if the sessions were in fact on different days. Additionally, was the amount of time between sessions controlled? What was the average time between sessions? More information needs to be recorded concerning this portion of the Methods section.

Line 139: The sentence starting on this line is winding and confusing. Consider:

"This first VR system (VR_sysI) replicates straight-line biomechanically but does not provide inertial compensation for simulating turns. We hypothesize that when linear inertia is compensated, rotational inertia can be neglected and the participant’s perception of turning can be induced using visual cues."

Line 145: "It is worth stating that all the participants recruited were included in data analyses of the VR systems they had tried during the main sessions." Does this indicate that some participants were eventually excluded? Reference Table 5 for clarity.

Line147: What is considered "a few" subjects? Was this a statistically motivated decision?

Line 159: "harder"

Line 162: Participant recruitment was restarted? Add reference to Table 5 for clarity.

Line 175: Former participants across all previous trials? Add reference to Table 5 for clarity.

Line 179: Formatting issues.

Line 182: Figure 2 should be placed further up to provide some clarification.

Line 184: Remove "Some." Additionally, the note of * does not provide enough information. What does "may not be enough" mean? Additionally, intra-ocular distance being recorded should be mentioned further up in the Methods section under "Participants."

Line 212: I am delighted to see that you completed additional statistical analyses concerning participant size.

Line 229: This line answers some of the questions I have listed above. However, these answers should be presented sooner to prevent undue confusion.

Table 5: I would reference this table in the Participant section, as well as other lines noted above.

Table 7: I feel like a better way to lay out this table is to include the demographic data of the participants, and indicate # of training sessions by participant. Additionally, training sessions by environment/system would also be enlightening. Maybe Fig. 2 can be modified in some way to include this?

Line 277: A visible pattern that is not significant, is not significant. This should not be presented as such. This issue is repeated at line 312.

Line 399: I am not convinced that motion sickness tends to " throw the participant's concentration away." There is no citation for this, and your study does not investigate concentration as a DV.

Line 450: Here is the line I have been waiting for. This would be best being stated in the "Participant" section, and then reiterated here.

Line 461: I appreciate this acknowledgment that the "wow" factor may have worn off by this point.

Fig 2: Why are the participants not labeled as "Subject 1, 2" etc.? This graph makes it very difficult to quickly refer to individual participants. Additionally, your caption for this figure only partially explains what I am looking at. Having more labels would benefit this graph immensely. Is this not the same thing as Table 5?

Overall, I feel like this study would benefit from more revision. Your introduction was succinct, and would be strengthened by expanding on your literature stepping stones. Further information in the Methods section would also benefit this manuscript.

I understand this consists of three major parts and therefore will be more winding than a short experiment, but I found myself constantly backtracking to ensure I was properly digesting the content. I still do not fully understand how training sessions were organized. Did the participants recieve training sessions for each environment? What was the average time gap between training sessions? There is a possibility that time may impact the outcome as well. Additionally, the three VR environment/systems blended together heavily, to the point of confusion. You might fair better by clearly seperating these three environments into their own sections, with an emphasise on clarifying participation/recruitment and methods. Improving Fig. 2 to aid with this should also be a priority.

Reviewer #2: The manuscript is not technically sound. The results of this study rely on there being no difference between MSAQ and IPQ results between the VR systems used. If there were significant differences between these systems, then the data would mandate an entirely different set of analyses than what was used in this study. However, the authors state their comparison across VR training systems is underpowered which makes a comparison moot. The authors need to increase the sample size of this study so that their comparison across VR systems is not underpowered and they can have statistical confidence in their results As it stands, the underpowered result cannot be trusted as indicative one way or another that VR systems were no different in MSAQ and IPQ. Failing to reject the null hypothesis should not be taken as evidence of the null hypothesis. Without this additional data, the findings of this study are equivocal.

Additionally, the authors claim that training sessions reduced MSAQ scores. However, training sessions was not the only independent variable in the study. Exposure to VR system type is a second independent variable which was not intended by the authors. This study would benefit from a different methodology that evaluated only one type of VR system. As the study stands, it’s impossible to say if training session alone led to reduced MSAQ scores. Perhaps it was changing VR systems across participants which led to reduced MSAQ scores. Perhaps it was both VR system type exposure, and repeated exposures (training sessions) which led to the reduction in MSAQ scores. The methodology and underpowered comparison of VR systems MSAQ and IPQ scores makes this impossible to determine.

The statistical analyses have not been performed appropriately and rigorously. The samples are undersized, as stated by the authors, and the decision against using multiple comparisons is not compelling or rigorous.

Reviewer #3: This project examines the role of repeated exposure to a virtual environment in order to reduce occurrences of VIMS without sacrificing immersion or presence. The researchers exposed participants to a large immersive virtual environment that the navigated through using a wheelchair – relation between wheelchair control and perceptual information was altered across three conditions. Participants were exposed to up to four pre-experimental training sessions to mitigate VIMS. Once VIMS was reduced to ‘tolerable’ levels researchers were interested in whether the different control/information coupling paradigms influenced presence in the virtual environment.

This was a timely and interesting project that would be of interest to the VR community and would make a contribution to the literature. However, in trying to be concise the structure and process of the studies can be hard to follow. Authors may want to reorganize to first address the aspects of the study that are designed to mitigate VIMS, then separately discuss the aspects of the study that are designed to address the differences in the VR system mechanics. It seems there are two goals – reduce sickness and maintain level of presence – for clarity, may need to explicitly organize the introduction, methods, analysis, and discussion in this manner. While VIMS and presence are related, discussing them separately may make it easier for readers to follow.

In terms of the analysis, I indicated ‘no’ because it is not clear why non-parametric statistics are not used across the board as your research questions/goals seem to suggest “yes/no” rather than inferential answers. Nonparametric analyses are perfectly appropriate for the questions you were addressing (particularly since the MANOVA did not reveal significant difference – which was the hope of the researchers?).

Minor things:

Just check for typos and tense/number agreement in sentences

For figure 6 – is the reason why the three systems are presented out of order? If so, should be noted in the caption.

For the word cloud figures – in the captions need to indicate what the reader should take note of – does the relative ‘strength’ of particular terms support your hypotheses?

6. PLOS authors have the option to publish the peer review history of their article (what does this mean?). If published, this will include your full peer review and any attached files.

Reviewer #1: No

Reviewer #2: **Yes: **Justin Munafo

Reviewer #3: **Yes: **L. James Smart Jr.

---

## [Author Response · Author response to Decision Letter 0]

17 Apr 2021

Comments Responses

Reviewer #1: 

Overall: Overall, I feel like this study would benefit from more revision. Your introduction was succinct, and would be strengthened by expanding on your literature steppingstones. Further information in the Methods section would also benefit this manuscript. 

I understand this consists of three major parts and therefore will be more winding than a short experiment, but I found myself constantly backtracking to ensure I was properly digesting the content. I still do not fully understand how training sessions were organized. Did the participants receive training sessions for each environment? What was the average time gap between training sessions? There is a possibility that time may impact the outcome as well. Additionally, the three VR environment/systems blended together heavily, to the point of confusion. You might fair better by clearly seperating these three environments into their own sections, with an emphasise on clarifying participation/recruitment and methods. Improving Fig. 2 to aid with this should also be a priority. 

A. We tried to apply all of these comments as addressed below. Thank you for the constructive comments.

Literature review is expanded.

More information is added to the Methods section.

Fig.2 depicts which systems were tried during the training and the main sessions.

An explanation is now added to Methods/Experimental procedure to provide more details on training sessions, as:

“During the training sessions, participants were simply exposed to different VR scenes and asked to freely “move around” as long as they feel like it. The training sessions were finished, however, whenever the participant asked for it. The duration of these sessions was between 5 to 30 minutes”.

The average time gap was 8.1 days. This is now added to the manuscript.

This manuscript is a story on continuous wheelchair virtual reality development with the users' comments being considered during this continuous development. This kind of development allows fast consideration of user's perception and faster iterations between VR systems. Separating every section into two or three parts could help with understanding each part better but will harm the flow of the “story”, making the logic of transitions between the sections harder to follow. Also, discussion on IPQ and MSAQ results are intertwined and there are several parts in the Discussion where it applies to the experiment as a whole, and not just to MSAQ or IPQ, such as the discussion on the sample size. Hence, with all due respect, we decided to keep the Methods as is. However, we did add subtitles of IPQ and MSAQ (next heading: training sessions/main sessions) to the Results and Discussion sections to organize the information.

Fig. 2 is improved now. Also, another figure (Fig 3) is added to the paper to help elaborate the ambiguities of the experimental procedure.

1- Line 34: I'm confused. VR system or environments? System reads as the physical hardware, the actual system that the VR environment is created by. 

A. Thank you for pointing out the unclarity. We changed the wordings and added an explanation early in the manuscript to clarify that there was a VR environment with three approaches to replicating rotational inertia (using three VR systems). 

“Three different approaches were taken to simulate wheelchair maneuvers in the VR environment, using three VR systems. To be brief, here we call the VR environment actuated using each approach a “VR_sys”.

2- Line 18: The users' sense of presence is a major focus of this paper. Defining exactly what that is would strengthen your paper immensely. Is this the sense of immersion? (I see in line 80 that these are listed as two separate things.) Please define this term. 

A. Immersion affects presence but is a separate thing. Immersion is objective and deals with the level of sensory fidelity provided by a VR system, but presence is subjective and may vary from one user to the other in a given VR environment. The definition of VR is added to the introduction as:

“Presence is the perception of transportation to the virtual scene and feeling as being there”.

3- Line 54: Missing period after [8]. 

A. Corrected.

4- Line 58: It is "stimuli."

A. Corrected.

5- Line 65: VR environment again, verses system. This is later changed in the Materials section (line 134). If these two are the same thing, it would be best to choose a single term. If there is some major difference between "environment" vs "system," that terminology should be established in the Introduction section. 

A. We changed the wordings and added an explanation early in the manuscript to clarify that there was a VR environment with three approaches to replicating rotational inertia (using three VR systems).

6- Line 89: Why four? I would greatly appreciate an explanation in this paragraph about why this number was chosen, as well as any citations that may support this.

A. The explanation you are asking for was in the method section, which is now displaced to Introduction/This study. That is:

“This protocol was designed based on research studies that have reported that having participants trying VR in four [12], five[13], [19] and six [3] sessions had helped them acclimatized to motion sickness. These training sessions should be held on different days [8], as sleeping between the sessions helps to promote neuro-plasticity (repairing and forming new connections in the nervous system).” 

7- Line 89: Numbers under 10 (that are not reference numbers) are spelled out.

A. Corrected here and elsewhere in the manuscript.

8- Line 102: What was the exclusion criteria? Motion sickness can also be affected by a history of concussion, TBI, etc. What does "able-bodied" actually mean?

A. Being able-bodied is used in contrast to having disabilities. Specifically, in here, being independent of using a wheelchair was meant. Able-bodied is now explained in the manuscript.

Participants were excluded if they had a musculoskeletal injury that affects normal wheelchair use, exercise-induced asthma, or heart disease. Their physical readiness was assessed before the experiments using ParQ and You physical readiness questionnaire. Other conditions not included in our exclusion criteria were accepted to affect the initial state of the participants with regard to their susceptibility to motion sickness. Our objective was to show no matter what the initial state was, the participant would be ready to undertake the experiments after a maximum of four conditioning sessions, based on their own judgment.

Other exclusion criteria were:

Neuromuscular condition e.g. multiple sclerosis, motor neuron disease

Pre-existing injury or pain during exertion in upper extremities by using PAR-Q questionnaire

Prescribed drugs for neuro-musculoskeletal pain or which have related side-effects

9- Line 109: Again, with spelling out numbers under 10 that are not references. Make sure this is consistent throughout the manuscript. 

A. Corrected. We made sure it is consistent throughout the manuscript.

 10- Line 110: I did not initially realize that the sessions were held on different days. This should be clarified sooner, as the past tense used in line 117 confused me to if the sessions were in fact on different days. Additionally, was the amount of time between sessions controlled? What was the average time between sessions? More information needs to be recorded concerning this portion of the Methods section. 

A. This paragraph is now placed further up in the manuscript, where we explain our study in the Introduction.

The conditioning sessions were only required to be held on different days, so the exact time between these sessions was not recorded. However, the time between those sessions was usually about 1 day to 1 week.

11- Line 139: The sentence starting on this line is winding and confusing. Consider: "This first VR system (VR_sysI) replicates straight-line biomechanically but does not provide inertial compensation for simulating turns. We hypothesize that when linear inertia is compensated, rotational inertia can be neglected and the participant’s perception of turning can be induced using visual cues."

A. Applied as suggested. Thank you for the help.

12- Line 145: "It is worth stating that all the participants recruited were included in data analyses of the VR systems they had tried during the main sessions." Does this indicate that some participants were eventually excluded? Reference Table 5 for clarity.

A. No. No data were excluded from the analyses. For resolving the ambiguity, the sentence was re-written as:

“It is worth stating that since not all the participants tried the same VR systems in their experiments, data of each participant was included in the analysis of the system(s) that they had tried”.

13- Line147: What is considered "a few" subjects? Was this a statistically motivated decision?

A. Three subjects, to be exact. This is depicted in Fig. 2 (reference added to the text).

No. The statistical analyses were completed after the completion of the experiments. As stated in the manuscript, this study had an iterative approach, seeking to reach a motion-sickness-free and representing VR system for wheelchair users. Thus, we made changes based on the participants’ comments and thereby made VR_sysII and VR_sysIII.

14- Line 159: "harder"

A. Corrected.

15- Line 162: Participant recruitment was restarted? Add reference to Table 5 for clarity.

A. “restarted” was substituted by “resumed”. Thank you for pointing it out.

16- Line 175: Former participants across all previous trials? Add reference to Table 5 for clarity.

A. Every participant that had completed two main sessions. Also, one participant declined from completing the third session. There was a reference to Fig 2 right after that sentence, which shows which participants completed the third main session, so reference to that table was not added to this sentence.

17- Line 179: Formatting issues.

A. This sentence was re-written to address your comment.

18- Line 182: Figure 2 should be placed further up to provide some clarification.

A. Applied.

19- Line 184: Remove "Some." Additionally, the note of * does not provide enough information. What does "may not be enough" mean? Additionally, intra-ocular distance being recorded should be mentioned further up in the Methods section under "Participants."

A. “Some” is removed.

The method of recording intra-ocular distance is now added to the Participants section.

The note of * is also updated as: “May not be enough if there are fast-moving objects in the scene that are not controlled by the user”.

20- Line 212: I am delighted to see that you completed additional statistical analyses concerning participant size.

A. We are happy to hear that.

21- Line 229: This line answers some of the questions I have listed above. However, these answers should be presented sooner to prevent undue confusion.

A. The message of this paragraph is now added to the Introduction section. 

22- Table 5: I would reference this table in the Participant section, as well as other lines noted above.

A. We are afraid the reference to that table in the Participants section is not practical, as it needs the IPQ and MSAQ to be defined and discussed first. However, reference to Fig 2 is now added to Participants section, as it represents similar information. Other notes above are also applied.

23- Table 7: I feel like a better way to lay out this table is to include the demographic data of the participants, and indicate # of training sessions by participant. Additionally, training sessions by environment/system would also be enlightening. Maybe Fig. 2 can be modified in some way to include this?

A. Table 7 is now modified to include demographic data next to # of training sessions.

Training sessions by environment/system would also be enlightening is already included in Fig 2.

24- Line 277: A visible pattern that is not significant, is not significant. This should not be presented as such. This issue is repeated at line 312.

A. The sentences are re-written as below to address this problem:

- Although P and S did not show statistically significant results, the downward trend in these motion sickness categories, as it is depicted in Fig 4, suggests potential clinical relevance.

- … we see a considerable difference between MSAQ scores of VR_sysII and the other systems which is indicative of potential clinical impact, but more research is needed to conclude this.

25- Line 399: I am not convinced that motion sickness tends to " throw the participant's concentration away." There is no citation for this, and your study does not investigate concentration as a DV.

A. With all due respect, there are references in the literature for this. Two references are added to the text now.

26- Line 450: Here is the line I have been waiting for. This would be best being stated in the "Participant" section, and then reiterated here. 

A. This sentence is stated in the Participants section too, now. Thank you for the comment.

27- Line 461: I appreciate this acknowledgment that the "wow" factor may have worn off by this point.

A. Happy to hear that.

28- Fig 2: Why are the participants not labeled as "Subject 1, 2" etc.? This graph makes it very difficult to quickly refer to individual participants. Additionally, your caption for this figure only partially explains what I am looking at. Having more labels would benefit this graph immensely. Is this not the same thing as Table 5?

A. Labels of the participants are now added to Fig 2. The caption is also expanded. 

This graph has an overlap with what that table (now Table 3) and Table 4 represent, as those tables show what data is available for each questionnaire, while this graph shows what systems each participant was exposed to, and in which sessions. Also, it shows how many training sessions each participant needed.

Reviewer #2: 

Overall: The manuscript is not technically sound.…

The statistical analyses have not been performed appropriately and rigorously. The samples are undersized, as stated by the authors, and the decision against using multiple comparisons is not compelling or rigorous.

A. We are sorry you felt that there were technical problems with this manuscript, and we hope that with the explanations provided, you feel more confident about the quality of this manuscript.

With all due respect, we disagree with statistical analyses not being performed appropriately and rigorously. Although the samples are undersized, every effort was made to add to the power of the analyses and that is why instead of using non-parametric methods that perfectly fitted our research questions, we used parametric methods wherever possible so that the power of the analyses are boosted. The statistical grounds for every decision made for which analyses to be used is presented in Tables 4 to 6. 

29- The results of this study rely on there being no difference between MSAQ and IPQ results between the VR systems used. If there were significant differences between these systems, then the data would mandate an entirely different set of analyses than what was used in this study.

A. We are afraid we do not follow your logic here. We were interested to see, among others, if there were any differences among the VR systems for IPQ and MSAQ and we ran MANOVA hypothesis testing for it. Every hypothesis test will eventually have a “maintain” or” reject” result for the null hypothesis. Why would a “maintain” result harm our analysis? 

30- However, the authors state their comparison across VR training systems is underpowered which makes a comparison moot. The authors need to increase the sample size of this study so that their comparison across VR systems is not underpowered and they can have statistical confidence in their results As it stands, the underpowered result cannot be trusted as indicative one way or another that VR systems were no different in MSAQ and IPQ.

A. We were not trying to show that there was no difference between the systems regarding MSAQ or IPQ, or the systems were “similar”. In contrast, we were interested to see which system is the best among the three systems, regarding motion sickness or presence. This analysis, however, did not show any difference among the systems, but this was not the only outcome of this study, nor the most important one. In this study we could show, with statistical significance, that: 

- A maximum of 4 training sessions indeed decrease the motion sickness to a tolerable level with a total observed power of 0.91 and a total effect size of 0.72 (for 3 of 5 subcategories, plus the explicit declaration of the participants), 

- About 20% of the variability in involvement and experienced realism scores of presence is accounted for by the VR system.

- Inverse relation between MSAQ and IPQ: The more the person suffers from motion sickness, the less they grade their presence in VR.

Despite that, as emphasized in the manuscript, “The statistics were used with great care, but they were not the main outcome or focus of this work. Rather, this study was about the development of three system iterations in time, and how people reacted to training on one or many of these systems”.

May we also add that we believe that research studies should not be published only on the ground of having statistically significant results, as this will bias the published literature to those research findings that have statistical significance. 

31- Failing to reject the null hypothesis should not be taken as evidence of the null hypothesis. Without this additional data, the findings of this study are equivocal.

A. That is true and we did not do it. As mentioned in the previous comment: We were not trying to show that there was no difference between the systems regarding MSAQ or IPQ, or the systems were “similar”. In contrast, we were interested to see which system is the best among the three systems, regarding motion sickness or presence.

32- Additionally, the authors claim that training sessions reduced MSAQ scores. However, training sessions was not the only independent variable in the study. Exposure to VR system type is a second independent variable which was not intended by the authors. This study would benefit from a different methodology that evaluated only one type of VR system. As the study stands, it’s impossible to say if training session alone led to reduced MSAQ scores. Perhaps it was changing VR systems across participants which led to reduced MSAQ scores. Perhaps it was both VR system type exposure, and repeated exposures (training sessions) which led to the reduction in MSAQ scores. The methodology and underpowered comparison of VR systems MSAQ and IPQ scores makes this impossible to determine.

A. This is correct that participants were exposed to different VR systems in the training sessions, but the conclusion remains sound, as we were concerned with exposure to VR as a whole and not with VR constraints or conditions: “exposure to VR for a maximum of four training sessions diminishes the motion sickness to a tolerable level, as indicated by the user, regardless of the type of the VR, provided that the sessions are held on different days.

Furthermore, Fig 2 shows that changing the exposure to VR system type was not common among all of the subjects and cannot be causing the reduction in motion sickness, as suggested by the reviewer.

33- In line 83 the author refers to fake movements in a user’s peripheral vision. This is unclear.

The author should provide an explanation of what a fake movement is or means.

A. This explanation is now added to the text: “movements that are inaccurate, flickering, encompass lags, etc.”.

34- In lines 159 the authors have a typo, “arder”

A. Corrected to “harder”

35- Unable to understand what the preconditions -3 to 0 trials are. The authors should provide more information about what users did during these training sessions.

A. This explanation is now added to the text for elaboration:

“Participants took 1 to 4 training sessions based on their needs. Since the number of the training sessions was different for different subjects and since the training sessions preceded the main sessions, the last training session was named 0 and the other training sessions were named retrospectively. This way, the first main session which comes after the last training session is named session 1 for all participants, while keeping the chronological sequence of all sessions for everyone”.

An explanation is now added to Methods/Experimental procedure to provide more details on training sessions, as:

“During the training sessions, participants were simply exposed to different VR scenes and asked to freely “move around” as long as they feel like it. The training sessions were finished, however, whenever the participant asked for it. The duration of these sessions was between 5 to 30 minutes”.

36- Lines 214-215 the authors say, “the statistical procedures are selected in a way to maximize the

power. Therefore, wherever possible, the parametrics methods used that have higher statistical power than non-parametric methods. However if the assumptions were not met, the

non-parametrics methods have been utilized.”

- This is confusing. The authors should be using analyses that meet the constraints of the

statistical assumptions. If this is what the authors did, I suggest rewriting this passage to

simplify it. The authors can simply say they used parametric and nonparametric tests when appropriate.

A. This paragraph is re-written to address this comment, by including your suggestion.

37- The justification for not using a Bonferonni correction, or any other statistical correction for

multiple comparisons, is untenable. The decision to correct for multiple comparisons cannot be made on the basis of what is convenient for the study or results. The authors state that

because they’re looking for similarities as opposed to differences, corrections for multiple comparisons aren’t necessary. The opposite is true. The authors should use conventional statistical methods (i.e. multiple comparisons corrections) which adhere to widely accepted and established statistical practices to demonstrate any claims they make based on their data.

A. We apologize that this paragraph is mistakenly put in this paper and actually belongs to (and is an excerpt of) another paper we have already published from the same project (reference 22 of this manuscript). The argument made was for that publication. This paragraph is deleted altogether. Thank you for pointing it out.

38- The authors should report the mean and standard deviation for MSAQ scores, and should report the sample size, the p value, and F value for the MANOVA.

A. Means and standard deviations for MSAQ scores were already included in Fig 5 and Table 8. Also, sample size and P-values were already reported in Tables 4 to 6. F values are now added to Table 4.

39- Lines 288-294 editorialize in the results section. The authors should plainly state their result,

and expand upon the importance of the result in the discussion section.

A. All editorial information including the section you indicated are now moved to Discussion.

40- Line 445-447 The authors state that some results of the study could not be confirmed statistically. However, a result cannot exist without empirical evidence. The authors claim here that a result exists in which they have no evidence for. The authors should remove this sentence.

A. That sentence is re-written now as: “some of the differences detected in this study could not be confirmed statistically”.

41 Lines 414-425 The authors state that there was no significant difference in MSAQ or IPQ for any VR system, but then state that a larger sample size would be needed to identify significant results. The authors then claim that future work doesn’t need a larger sample size. This is problematic for two reasons. The authors specifically identify that their null result in MSAQ and IPQ across systems is likely due to too small a sample size. Because this is the case, the authors should not claim there is or is not a difference between VR training systems. A second concern is that the authors say future work should not focus on getting a large enough sample size to identify statistical differences between the systems. However, this would leave future researchers in the same bind the current authors are in; their analyses are underpowered and cannot be used to validate their claims one way or another.

A. As you said, we have recognized in our manuscript that a larger sample size, if implanted, could have probably shown some differences between the systems, which by the way, is not the main outcome of this study. Also, we have said: “future work should focus on creating a VR user experience that is a significant improvement on the designs used in this study, rather than recruiting a larger sample of participants, simply to demonstrate the differences between these systems”. There is no controversy between these two sentences, and they are not problematic, as in the latter one we are saying, in other words, that although larger sample size could have “saved” our data, our main objective was a fast iteration between the systems to come with a better VR system. Then we have shown the future avenue by identifying that non of these VR systems are “perfect” and future studies should not just repeat these tests with (one of) these VR systems using larger sample sizes, merely to show a “significant difference” between these systems. Rather, they need to come with an even better VR system and validate it using an appropriate statistical procedure. 

With regard to recognizing differences among systems, we would like to emphasize again that we never claimed differences between the systems more than what participants' comments, shown in the form of word clouds, tells us (with no statistical judgment on that). The take-home messages of this paper are: 

-The 3 VR systems showed relatively high presence and low motion sickness.

- Up to four training sessions is effective in mitigating motion sickness.

- Presence and motion sickness are inversely related.

- About 20% of the variability in involvement and experienced realism scores of presence is accounted for by the VR system (statistically significant).

Reviewer #3: 

Overall: This project examines the role of repeated exposure to a virtual environment in order to reduce occurrences of VIMS without sacrificing immersion or presence. The researchers exposed participants to a large immersive virtual environment that the navigated through using a wheelchair – relation between wheelchair control and perceptual information was altered across three conditions. Participants were exposed to up to four pre-experimental training sessions to mitigate VIMS. Once VIMS was reduced to ‘tolerable’ levels researchers were interested in whether the different control/information coupling paradigms influenced presence in the virtual environment.

This was a timely and interesting project that would be of interest to the VR community and would make a contribution to the literature. However, in trying to be concise the structure and process of the studies can be hard to follow. Authors may want to reorganize to first address the aspects of the study that are designed to mitigate VIMS, then separately discuss the aspects of the study that are designed to address the differences in the VR system mechanics. It seems there are two goals – reduce sickness and maintain level of presence – for clarity, may need to explicitly organize the introduction, methods, analysis, and discussion in this manner. While VIMS and presence are related, discussing them separately may make it easier for readers to follow. 

A. Thank you for the positive view on our manuscript.

This manuscript is a story on continuous wheelchair virtual reality development with the users' comments being considered during this continuous development. This kind of development allows fast consideration of user's perception and faster iterations between VR systems. Separating every section to two or three parts could help with understanding each part better but will harm the flow of the “story”, making the logic of transitions between the sections harder to follow. Also, discussion on IPQ and MSAQ results are intertwined and there are several parts in the Discussion where it applies to the experiment as a whole, and not just to MSAQ or IPQ, such as the discussion on the sample size. Hence, with all due respect, we decided to keep the Methods as is. However, we did add subtitles of IPQ and MSAQ (next heading: training sessions/main sessions) to the Results and Discussion sections to organize the information.

42- In terms of the analysis, I indicated ‘no’ because it is not clear why non-parametric statistics are not used across the board as your research questions/goals seem to suggest “yes/no” rather than inferential answers. Nonparametric analyses are perfectly appropriate for the questions you were addressing (particularly since the MANOVA did not reveal significant difference – which was the hope of the researchers?).

A. The parametric methods are more powerful than non-parametric methods. On the other hand, small sample sizes reduce the power of a study. Since the power of this study for our research questions (that involve high between-subject variations) is fragile, we used every effort to add to the power of the analyses. Thus, where possible, parametric methods were used.

This explanation is now added to the Methods/Statistical procedure.

43- Just check for typos and tense/number agreement in sentences

A. The whole manuscript was checked for grammatical errors and typos. 

44 For figure 6 – is there a reason why the three systems are presented out of order? If so, should be noted in the caption.

A. Yes. VR systems were named chronologically, but the experiments revealed that VR_sysII and VR_sysIII received the lowest and the highest presence scores, respectively. To assess what portion of this difference was accounted for by the VR system, a regression analysis was performed with having VR_sysII at the far left and VR_sysIII at the far right of the axis, leading to VR_sysI being in the middle.

An explanation was added to the caption.

45- For the word cloud figures – in the captions need to indicate what the reader should take note of – does the relative ‘strength’ of particular terms support your hypotheses?

A. To avoid repeating the same relatively long sentence in the caption of the three word-clouds, this sentence was added to the text referring to those figures:

“In these figures, the size of each word is related to the rank of its repetition in the participants’ comments. Thus bigger words indirectly show that those words are concerns to the participants”.

---

## [Decision Letter · Decision Letter 1]

17 May 2021

PONE-D-21-03454R1

Motion sickness and sense of presence in a virtual reality environment developed for manual wheelchair users, with three different approaches

PLOS ONE

Dear Dr. Salimi,

Thank you for submitting your manuscript to PLOS ONE. After careful consideration, we feel that it has merit but does not fully meet PLOS ONE’s publication criteria as it currently stands. Therefore, we invite you to submit a revised version of the manuscript that addresses the points raised during the review process.

Reviewer 2 is satisfied with your changes. However, you will see that Reviewer 3 feels that considerable additional revision is needed. I agree with Reviewer 3; the requested changes seem reasonable, and will significantly enhance the value of your contribution.

We look forward to receiving your revised manuscript.

Kind regards,

Thomas A Stoffregen, PhD

Academic Editor

PLOS ONE

Reviewers' comments:

Reviewer's Responses to Questions

**Comments to the Author**

1. If the authors have adequately addressed your comments raised in a previous round of review and you feel that this manuscript is now acceptable for publication, you may indicate that here to bypass the “Comments to the Author” section, enter your conflict of interest statement in the “Confidential to Editor” section, and submit your "Accept" recommendation.

Reviewer #2: All comments have been addressed

Reviewer #3: (No Response)

2. Is the manuscript technically sound, and do the data support the conclusions?

Reviewer #2: Yes

Reviewer #3: Partly

3. Has the statistical analysis been performed appropriately and rigorously? 

Reviewer #2: Yes

Reviewer #3: No

4. Have the authors made all data underlying the findings in their manuscript fully available?

Reviewer #2: Yes

Reviewer #3: Yes

5. Is the manuscript presented in an intelligible fashion and written in standard English?

Reviewer #2: Yes

Reviewer #3: Yes

6. Review Comments to the Author

Reviewer #2: I want to thank the author for responding to/incorporating my feedback. I still think the underpowered analysis is problematic, but not fatal to the paper.

Reviewer #3: Thank you for the revisions you made and providing a rationale for the items that you left from the original submission. Unfortunately there are still some significant items that need to be addressed in the introduction and in your results/analysis.

You introduction is a general statement of the problem of VIMS in VR followed by a research example prior to discussing your work in the manuscript. The issue that the discussion of Chattha et al does not motivate or justify your manipulation/method, it is a jarring switch between the problem set-up and the overview of your project. You would be better served to address why movement/experience using the wheel chair is important (in general the connection between perception and action). In addition your stated goals for the project (process of developing a more viable VR interaction) really don't map onto the the analyses you propose. Again you state (as in the original) that the statistical outcomes were not the goal, rather the development of a compelling and sickness "free" virtual experience. Given this it is hard to understand the choice of analyses (more power does not help when the choice of analysis itself is non-appropriate for the stated goal). It appears that you are asking for a perceptually ranked ordering of the VR systems, again non-parametrics are tailored for this type of analysis. MANOVA is not designed to tell 'optimum' or 'best', just differences and it is not a 'solution' for dealing with small samples sizes (in fact it works much better with larger samples). Adjusting the frame of your introduction and analysis may solve these issues.

7. PLOS authors have the option to publish the peer review history of their article (what does this mean?). If published, this will include your full peer review and any attached files.

Reviewer #2: No

Reviewer #3: **Yes: **L. James Smart Jr.

---

## [Author Response · Author response to Decision Letter 1]

17 Jun 2021

Reviewer #2 

Comment: I want to thank the author for responding to/incorporating my feedback. I still think the underpowered analysis is problematic, but not fatal to the paper. 

Answer: We are happy to read that you feel more confident about the paper.

Reviewer #3 

Overall: Thank you for the revisions you made and providing a rationale for the items that you left from the original submission. Unfortunately, there are still some significant items that need to be addressed in the introduction and in your results/analysis. 

Answer: We are happy to address them. Thank you for your constructive approach. 

1: Your introduction is a general statement of the problem of VIMS in VR followed by a research example prior to discussing your work in the manuscript. The issue that the discussion of Chattha et al does not motivate or justify your manipulation/method, it is a jarring switch between the problem set-up and the overview of your project. You would be better served to address why movement/experience using the wheel chair is important (in general the connection between perception and action). 

Answer: The Chattha paragraph was relocated to a more proper place (where the factors triggering motion sickness was discussed). In addition, a paragraph was added, as suggested by the reviewer, before switching to discussing our study, to facilitate a better transition:

“As mentioned, when there is a mismatch between vestibular and visual stimuli, VIMS could trigger. This is roughly whenever the user moves around, or looks around, in the virtual world. We also mentioned that VR has a potential to be used to assist with many different areas, including for training and conducting related research studies for wheelchair users, as one important part of the society who face miscellaneous difficulties on a daily basis; difficulties ranging from troubles in moving around and navigating in and outside home and accessing buildings, to secondary injuries that they usually undergo as a consequence of the above-normal and repetitive load they experience in their upper-body which they have to use for ambulation. We developed this study to simulate navigation using wheelchair in the VR, and since combination of navigation and VR has a high probability of inducing VIMS, we concentrated on mitigating VIMS while trying to ascertain higher sense of presence.”

2: In addition, your stated goals for the project (process of developing a more viable VR interaction) really don't map onto the the analyses you propose. Again, you state (as in the original) that the statistical outcomes were not the goal, rather the development of a compelling and sickness "free" virtual experience. Given this it is hard to understand the choice of analyses (more power does not help when the choice of analysis itself is non-appropriate for the stated goal). It appears that you are asking for a perceptually ranked ordering of the VR systems, again non-parametrics are tailored for this type of analysis. MANOVA is not designed to tell 'optimum' or 'best', just differences and it is not a 'solution' for dealing with small samples sizes (in fact it works much better with larger samples). Adjusting the frame of your introduction and analysis may solve these issues.

Answer: We understand that the wording in the literature, “statistics are not the main focus of the study” has made you uncertain about why then we have made many difficult statistical roundabouts to deal with the data. Thus, we have revised that paragraph and tried to explain our approach better in the introduction, as below:

“We also wish to note that although Statistics was not the focus of this research study, the statistical approaches were selected with care rather than to simplify the analyses. In order to get the most rigorous results, we carefully assessed each outcome for the six analyses to determine which of the parametric or nonparametric methods would suit the conditions of that dataset and chose the proper statistical approach accordingly”.

In case you still are not very sure about the rational behind choosing the statistical approaches in this manuscript, we have provided more explanations in the following:

It seems that the point you are raising is more about “types of statistical analysis” e.g., difference-finding, similarity-finding, or optimum finding, than using parametric or non-parametric approaches, because you can perform all those analyses using both parametric and non-parametric methods, where conditions (e.g., normal distribution) are met. In this manuscript, we have used both methods, where appropriate, to see if the systems are different, and if yes, which of the 3 systems are better than the others; a normal and very frequent use of MANOVA in the literature.

As reported frequently in the literature, e.g., Hopkins (2018), parametric methods should be used when we have continuous data that is normally distributed, and non-parametric methods should be used when we have categorical data, or we do not have normal distribution (exactly what we did in this manuscript). Despite what you are suggesting, it is never advised in the literature to use nonparametric methods when we have normal distribution (mainly our case).

I am not getting different type of the results from the two methods. Even with the non-parametric method, Kruskal-Wallis, I report the significance level, and based on that, I judge if there was a meaningful difference- just as what we did with the MANOVA.

Both parametric and non-parametric methods work better with larger sample sizes. That is true. But for a given fragile sample size, and when I get the same information from running MANOVA or its non-parametric counterpart: Kruskal-Wallis, I would prefer to use the parametric method rather than use a less powerful method that does a similar job. 

Finally, mixing parametric and non-parametric methods is not unprecedented in the literature. For example, the following has used both parametric and nonparametric tests in the same study, as per assumptions were satisfied, just like what we did in our study:

“Niksirat, K.S., Silpasuwanchai, C., Ahmed, M.M.H., Cheng, P., Ren, X., 2017. A framework for interactive mindfulness meditation using attention-regulation process. Conf. Hum. Factors Comput. Syst. - Proc. 2017-May, 2672–2684. https://doi.org/10.1145/3025453.3025914"

With all due respect, the authors of this manuscript understand that while using the non-parametric method for all the analyses of this manuscript would keep things neat, the more rigorous way is to use parametric methods wherever the conditions are met. This strengthens the integrity of the analysis and reduces the chance of type II errors. 

Footnote: Hopkins S, Dettori JR, Chapman JR. Parametric and Nonparametric Tests in Spine Research: Why Do They Matter? Global Spine Journal. 2018;8(6):652-654. doi:10.1177/2192568218782679

---

## [Decision Letter · Decision Letter 2]

27 Jul 2021

Motion sickness and sense of presence in a virtual reality environment developed for manual wheelchair users, with three different approaches

PONE-D-21-03454R2

Dear Dr. Salimi,

We’re pleased to inform you that your manuscript has been judged scientifically suitable for publication and will be formally accepted for publication once it meets all outstanding technical requirements.

Reviewer 2 was willing to accept your first revision, and Reviewer 3 recommends acceptance of the current manuscript; hence my decision to accept.

Kind regards,

Thomas A Stoffregen, PhD

Academic Editor

PLOS ONE

Additional Editor Comments (optional):

Reviewers' comments:

Reviewer's Responses to Questions

**Comments to the Author**

1. If the authors have adequately addressed your comments raised in a previous round of review and you feel that this manuscript is now acceptable for publication, you may indicate that here to bypass the “Comments to the Author” section, enter your conflict of interest statement in the “Confidential to Editor” section, and submit your "Accept" recommendation.

Reviewer #3: All comments have been addressed

2. Is the manuscript technically sound, and do the data support the conclusions?

Reviewer #3: Yes

3. Has the statistical analysis been performed appropriately and rigorously? 

Reviewer #3: Yes

4. Have the authors made all data underlying the findings in their manuscript fully available?

Reviewer #3: Yes

5. Is the manuscript presented in an intelligible fashion and written in standard English?

Reviewer #3: Yes

6. Review Comments to the Author

Reviewer #3: Thank you for making the adjustments to the introduction as requested. While I am still not 100% sure that that your analysis matches your research goal - I appreciate that you checked the data using both para and nonparametric methods to ensure that your outcomes were consistent. So I am ok with your explanation

7. PLOS authors have the option to publish the peer review history of their article (what does this mean?). If published, this will include your full peer review and any attached files.

Reviewer #3: **Yes: **L. James Smart Jr.

---

## [Editor Report · Acceptance letter]

2 Aug 2021

PONE-D-21-03454R2 

Motion sickness and sense of presence in a virtual reality environment developed for manual wheelchair users, with three different approaches 

Dear Dr. Salimi:

I'm pleased to inform you that your manuscript has been deemed suitable for publication in PLOS ONE. Congratulations! Your manuscript is now with our production department. 

Kind regards, 

on behalf of

Dr. Thomas A Stoffregen 

Academic Editor

PLOS ONE